# Training Subset Selection for Weak Supervision

**Hunter Lang**
MIT CSAIL
hjl@mit.edu

**Aravindan Vijayaraghavan**
Northwestern University
aravindv@northwestern.edu

**David Sontag**
MIT CSAIL
dsontag@mit.edu

## Abstract

Existing weak supervision approaches use all the data covered by weak signals to train a classifier. We show both theoretically and empirically that this is not always optimal. Intuitively, there is a tradeoff between the amount of weakly-labeled data and the precision of the weak labels. We explore this tradeoff by combining pretrained data representations with the *cut statistic* [23] to select (hopefully) high-quality subsets of the weakly-labeled training data. Subset selection applies to any label model and classifier and is very simple to plug in to existing weak supervision pipelines, requiring just a few lines of code. We show our subset selection method improves the performance of weak supervision for a wide range of label models, classifiers, and datasets. Using *less* weakly-labeled data improves the accuracy of weak supervision pipelines by up to 19% (absolute) on benchmark tasks.

## 1 Introduction

Due to the difficulty of hand-labeling large amounts of training data, an increasing share of models are trained with *weak supervision* [30, 28]. Weak supervision uses expert-defined "labeling functions" to programatically label a large amount of training data with minimal human effort. This *pseudo*-labeled training data is used to train a classifier (e.g., a deep neural network) as if it were hand-labeled data.

Labeling functions are often simple, coarse rules, so the pseudolabels derived from them are not always correct. There is an intuitive tradeoff between the *coverage* of the pseudolabels (how much pseudolabeled data do we use for training?) and the *precision* on the covered set (how accurate are the pseudolabels that we do use?). Using all the pseudolabeled training data ensures the best possible generalization to the population pseudolabeling function $\hat{Y}(X)$. On the other hand, if we can select a high-quality subset of the pseudolabeled data, then our training labels $\hat{Y}(X)$ are closer to the true label $Y$, but the smaller training set may hurt generalization. However, existing weak supervision approaches such as Snorkel [28], MeTaL [29], FlyingSquid [10], and Adversarial Label Learning [3] use *all* of the pseudolabeled data to train the classifier, and do not explore this tradeoff.

We present numerical experiments demonstrating that the status quo of using all the pseudolabeled data is nearly always suboptimal. Combining good pretrained representations with the *cut statistic* [23] for subset selection, we obtain subsets of the weakly-labeled training data where the weak labels are very accurate. By choosing examples with the same pseudolabel as many of their nearest neighbors in the representation, the cut statistic uses the representation's *geometry* to identify these accurate subsets without using any ground-truth labels. Using the smaller but higher-quality training sets selected by the cut statistic improves the accuracy of weak supervision pipelines by up to 19% accuracy (absolute). Subset selection applies to any "label model" (Snorkel, FlyingSquid, majority vote, etc.) and any classifier, since it is a modular, intermediate step between creation of the pseudolabeled training set and training. We conclude with a theoretical analysis of a special case of weak supervision where the precision/coverage tradeoff can be made precise.

36th Conference on Neural Information Processing Systems (NeurIPS 2022).

## 2   Background

The three components of a weak supervision pipeline are the *labeling functions*, the *label model*, and the *end model*. The labeling functions are maps $\Lambda_k : \mathcal{X} \to \mathcal{Y} \cup \{\varnothing\}$, where $\varnothing$ represents abstention. For example, for sentiment analysis, simple *token-based* labeling functions are effective, such as:

$$\Lambda_1(x) = \begin{cases} 1 & \text{``good''} \in x \\ \varnothing & \text{otherwise} \end{cases} \qquad \Lambda_2(x) = \begin{cases} -1 & \text{``bad''} \in x \\ \varnothing & \text{otherwise} \end{cases}$$

If the word "good" is in the input text $x$, labeling function $\Lambda_1$ outputs 1; likewise when "bad" $\in x$, $\Lambda_2$ outputs $-1$. Of course, an input text could contain both "good" and "bad", so $\Lambda_1$ and $\Lambda_2$ may conflict. Resolving these conflicts is the role of the *label model*.

Formally, the label model is a map $\hat{Y} : (\mathcal{Y} \cup \{\varnothing\})^K \to \mathcal{Y} \cup \{\varnothing\}$. That is, if we let $\mathbf{\Lambda}(x)$ refer to the vector $(\Lambda_1(x), \ldots, \Lambda_K(x))$, then $\hat{Y}(\mathbf{\Lambda}(x))$ is a single pseudolabel (or "weak label") derived from the vector of $K$ labeling function outputs. This resolves conflicts between the labeling functions. Note that we can also consider $\hat{Y}$ as a deterministic function of $X$. The simplest label model is *majority vote*, which outputs the most common label from the set of non-abstaining labeling functions:

$$\hat{Y}_{MV}(x) = \text{mode}(\{\Lambda_k(x) : \Lambda_k(x) \neq \varnothing\})$$

If all the labeling functions abstain (i.e., $\Lambda_k(x) = \varnothing$ for all $k$), then $\hat{Y}_{MV}(x) = \varnothing$. More sophisticated label models such as Snorkel [30] and FlyingSquid [10] parameterize $\hat{Y}$ to learn better aggregation rules, e.g. by accounting for the accuracy of different $\Lambda_k$'s or accounting for correlations between pairs $(\Lambda_j, \Lambda_k)$. These parameters are learned using unlabeled data only; the methods for doing so have a rich history dating back at least to Dawid and Skene [8]. Many label models (including Snorkel and its derivatives) output a "soft" pseudolabel, i.e., a distribution $\hat{P}[Y|\Lambda_1(X), \ldots, \Lambda_K(X)]$, and set the hard pseudolabel as $\hat{Y}(X) = \text{argmax}_y \, \hat{P}[Y = y|\Lambda_1(X), \ldots, \Lambda_K(X)]$.

Given an unlabeled sample $\{x_i\}_{i=1}^n$, the label model produces a pseudolabeled training set $\mathcal{T} = \{(x_i, \hat{Y}(x_i)) : \hat{Y}(x_i) \neq \varnothing\}$. The final step in the weak supervision pipeline is to use $\mathcal{T}$ like regular training data to train an *end model* (such as a deep neural network), minimizing the zero-one loss:

$$\hat{f} := \underset{f \in \mathcal{F}}{\text{argmin}} \, \frac{1}{|\mathcal{T}|} \sum_{i=1}^{|\mathcal{T}|} \mathbb{I}[f(x_i) \neq \hat{Y}(x_i)] \tag{1}$$

or a convex surrogate like cross-entropy. For many applications, we fine-tune a *pretrained* representation instead of training from scratch. For example, on text data, we can fine-tune a pretrained BERT model. We refer to the pretrained representation used by the end model as the *end model representation*, where applicable.

Notably, all existing methods use the full pseudolabeled training set $\mathcal{T}$ to train the end model. $\mathcal{T}$ consists of *all* points where $\hat{Y} \neq \varnothing$. In this work, we experiment with methods for choosing higher-quality subsets $\mathcal{T}' \subset \mathcal{T}$ and use $\mathcal{T}'$ in (1) instead of $\mathcal{T}$.

**Related Work.**   The idea of selecting a subset of high-quality training data for use in fully-supervised or semi-supervised learning algorithms has a long history. It is also referred to as *data pruning* [1], and a significant amount of work has focused on removing mislabeled examples to improve the training process [e.g., 26, 18, 7, 25]. These works do not consider the case where the pseudolabels come from deterministic labeling functions, and most try to estimate parameters of a specific noise process that is assumed to generate the pseudolabels. Many of these approaches require iterative learning or changes to the loss function, whereas typical weak supervision pipelines do one learning step and little or no loss correction. Maheshwari et al. [21] study *active* subset selection for weak supervision, obtaining a small number of human labels to boost performance.

In *self-training* [e.g., 34], an initial labeled training set is iteratively supplemented with the pseudolabeled examples where a trained model is most confident (according to the model's probability scores). The model is retrained on the new training set in each step. Yarowsky [40] used this approach starting from a *weakly*-labeled training set; Yu et al. [41], Karamanolakis et al. [15] also combine self-training with an initial weakly-labeled training set, and both have deep-model-based procedures for selecting confident data in each round. We view these weakly-supervised self-training methods as orthogonal to our approach, since their *main* focus is on making better use of the data that is not covered by weak rules, not on selecting good pseudolabeled subsets. Indeed, we show in Appendix

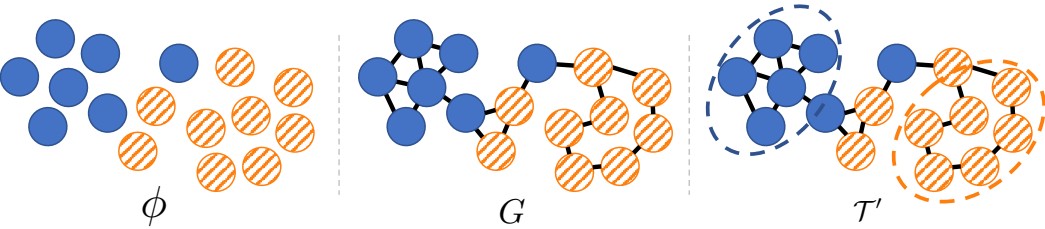

Figure 1: Cut statistic procedure. A representation $\phi$ is used to compute the nearest-neighbor graph $G$. Nodes that have the same pseudolabel as most of their neighbors are chosen for the subset $\mathcal{T}'$.

B.7 that combining our method with these approaches improves their performance. Other selection schemes, not based on model confidence, have also been investigated for self-training [e.g., 44, 24].

Muhlenbach et al. [23] introduced the *cut statistic* as a heuristic for identifying mislabeled examples in a training dataset. Li and Zhou [19] applied the cut statistic to self-training, using it to select high-quality pseudolabeled training data for each round. Zhang and Zhou [43] applied the cut statistic to co-training and used learning-with-noise results from Angluin and Laird [2] to optimize the amount of selected data in each round. Lang et al. [16] also used co-training and the cut statistic to co-train large language models such as GPT-3 [5] and T0 [31] with smaller models such as BERT [9] and RoBERTa [20]. These previous works showed that the cut statistic performs well in *iterative* algorithms such as self-training and co-training; we show that it works well in *one-step* weak supervision settings, and that it performs especially well when combined with modern pre-trained representations. Our empirical study shows that this combination is very effective at selecting good pseudolabeled training data *across a wide variety of label models, end models, and datasets*.

As detailed in Section 3, the performance of the cut statistic relies on a good representation of the input examples $x_i$ to find good subsets. Zhu et al. [45] also used representations to identify subsets of mislabeled labels and found that methods based on representations outperform methods based on model predictions alone. They use a different ranking method and do not evaluate in weakly-supervised settings. Chen et al. [6] also use pretrained representations to improve the performance of weak supervision. They created a new representation-aware label model that uses nearest-neighbors in the representation to label *more* data and also learns finer-grained label model parameters. In contrast, our approach applies to any label model, can be implemented in a few lines of code, and does not require representations from very large models like GPT-3 or CLIP. Combining the two approaches is an interesting direction for future work.

## 3 Subset Selection Methods for Weak Supervision

In this work, we study techniques for selecting high-quality subsets of the pseudolabeled training set $\mathcal{T}$. We consider two simple approaches to subset selection in this work: *entropy scoring* and the *cut statistic*. In both cases, we construct a subset $\mathcal{T}'$ by first ranking all the examples in $\mathcal{T}$, then selecting the top $\beta$ fraction according to the ranking. In our applications, $0 < \beta \leqslant 1$ is a hyperparameter tuned using a validation set. Hence, instead of $|\mathcal{T}|$ covered examples for training the end model in (1), we use $\beta|\mathcal{T}|$ examples. Instead of a single, global ranking, subset selection can easily be stratified to use multiple rankings. For example, if the true label balance $\mathbb{P}[Y]$ is known, we can use separate rankings for each set $\mathcal{T}_y = \{x_i : \hat{Y}(x_i) = y\}$ and select the top $\beta\mathbb{P}[Y = y]|\mathcal{T}|$ points from each $\mathcal{T}_y$. This matches the pseudolabel distribution on $\mathcal{T}'$ to the true marginal $\mathbb{P}[Y]$. For simplicity, we use a global ranking in this work, and our subset selection does not use $\mathbb{P}[Y]$ or any other information about the true labels. Below we give details for the entropy and cut statistic rankings.

**Entropy score.** Entropy scoring only applies to label models that output a "soft" pseudo-label $\hat{P}[Y|\mathbf{\Lambda}(X)]$. For this selection method, we rank examples by the Shannon entropy of the soft label, $H(\hat{P}[Y|\mathbf{\Lambda}(x_i)])$, and set $\mathcal{T}'$ to the $\beta|\mathcal{T}|$ examples with the lowest entropy. Intuitively, the label model is the "most confident" on the examples with the lowest entropy. If the label model is well-calibrated, the weak labels should be more accurate on these examples.

**Cut statistic [23].** Unlike the entropy score, which only relies on the soft label distribution $\hat{P}[Y|\mathbf{\Lambda}]$, the cut statistic relies on a good *representation* of the input examples $x_i$. Let $\phi$ be a representation for

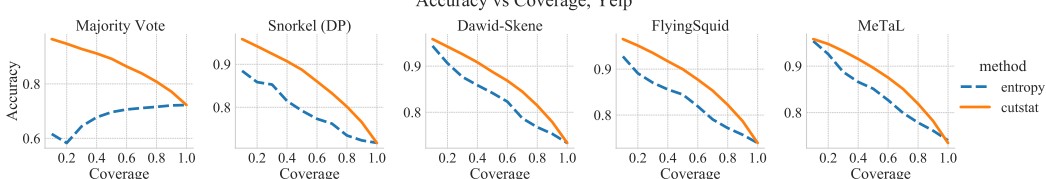

Figure 2: Accuracy of the pseudolabeled training set versus the selection fraction $\beta$ for five different label models. A pretrained BERT model is used as $\phi$ for the cut statistic. The accuracy of the weak training labels is better for $\beta < 1$, indicating that sub-selection can select higher-quality training sets.

examples in $\mathcal{X}$. For example, for text data, $\phi$ could be the hidden state of the `[CLS]` token in the last layer of a pretrained large language model.

Recall that $\mathcal{T} = \{(x_i, \hat{Y}(x_i)) : \hat{Y}(x_i) \neq \varnothing\}$. To compute the cut statistic using $\phi$, we first form a graph $G = (V, E)$ with one vertex for each covered $x_i$ and edges connecting vertices who are $K$-nearest neighbors in $\phi$. That is, for each example $x_i$ with $\hat{Y}(x_i) \neq \varnothing$, let

$$\text{NN}_\phi(x_i) = \{x_j : (x_i, x_j) \text{ are } K\text{-nearest-neighbors in } \phi\}.$$

Then we set $V = \{i : \hat{Y}(x_i) \neq \varnothing\}$, $E = \{(i, j) : x_i \in \text{NN}_\phi(x_j) \text{ or } x_j \in \text{NN}_\phi(x_i)\}$. For each node $i$, let $N(i) = \{j : (i, j) \in E\}$ denote its neighbors in $G$. We assign a weight $w_{ij}$ to each edge so that nodes closer together in $\phi$ have a higher edge weight: $w_{ij} = (1 + ||\phi(x_i) - \phi(x_j)||_2)^{-1}$. We say an edge $(i, j)$ is *cut* if $\hat{Y}(x_i) \neq \hat{Y}(x_j)$, and capture this with the indicator variable $I_{ij} := \mathbb{I}[\hat{Y}(x_i) \neq \hat{Y}(x_j)]$. As suggested in Figure 1, if $\phi$ is a good representation, nodes with few incident cut edges should have high-quality pseudolabels—these examples have the same label as most of their neighbors. On the other hand, nodes with a large number of cut edges likely correspond to mislabeled examples. The cut statistic heuristically quantifies this idea to produce a ranking.

Suppose (as a null hypothesis) that the labels $\hat{Y}$ were sampled i.i.d. from the marginal distribution $\mathbb{P}[\hat{Y} = y]$. Large deviations from the null should represent the most noise-free vertices. For each vertex $i$, consider the test statistic: $J_i = \sum_{j \in N(i)} w_{ij} I_{ij}$. The mean of $J_i$ under the null hypothesis is: $\mu_i = (1 - \mathbb{P}[\hat{Y}(x_i)]) \sum_{j \in N(i)} w_{ij}$, and the variance is: $\sigma_i^2 = \mathbb{P}[\hat{Y}(x_i)](1 - \mathbb{P}[\hat{Y}(x_i)]) \sum_{j \in N(j)} w_{ij}^2$. Then for each $i$ we can compute the Z-score $Z_i = \frac{J_i - \mu_i}{\sigma_i}$ and rank examples by $Z_i$. Lower is better, since nodes with the smallest $Z_i$ have the least noisy $\hat{Y}$ assignments in $\phi$. As with entropy scoring, we set $\mathcal{T}'$ to be the the $\beta|T|$ points with the smallest values of $Z_i$. We provide code for a simple ($<$ 30 lines) function to compute the $Z_i$ values given the representations $\{\phi(x_i) : x_i\}$ in Appendix C. Calling this function makes it very straightforward to incorporate the cut statistic in existing weak supervision pipelines. Since the cut statistic does not require soft pseudolabels, it can also be used for label models that only produce hard labels, and for label models such as Majority Vote, where the soft label tends to be badly miscalibrated.

## 3.1 Cut Statistic Selects Better Subsets

To explore the two scoring methods, we visualize how $\mathcal{T}'$ changes with $\beta$ for entropy scoring and the cut statistic. We used label models such as majority vote and Snorkel [30] to obtain soft labels $\hat{P}[Y|\mathbf{\Lambda}(x_i)]$, and set $\hat{Y}(x_i)$ to be the argmax of the soft label. We test using the **Yelp** dataset from the WRENCH weak supervision benchmark [42]. The task is sentiment analysis, and the eight labeling functions $\{\Lambda_1, \ldots, \Lambda_8\}$ consist of seven keyword-based rules and one third-party sentiment polarity model. For $\phi$ in the cut statistic, we used the `[CLS]` token representation of a pretrained BERT model. Section 4 contains more details on the datasets and the cut statistic setup.

For each $\beta \in \{0.1, 0.2, \ldots, 1.0\}$, Figure 2 plots the accuracy of the pseudolabels on the training subset $\mathcal{T}'(\beta)$. This shows how training subset quality varies with the selection fraction $\beta$. We can compute this accuracy because most of the WRENCH benchmark datasets also come with ground-truth labels $Y$ (even on the training set) for evaluation. Appendix B contains the same plot for several other WRENCH datasets and figures showing the histograms of the entropy scores and the $Z_i$ values.

Figure 2 shows that combining the cut statistic with a BERT representation selects better subsets than the entropy score for all five label models tested, especially for majority vote, where the entropy

scoring is badly miscalibrated. For a well-calibrated score, the subset accuracy should decrease as $\beta$ increases. These results suggest that the cut statistic is able to use the geometric information encoded in $\phi$ to select a more accurate subset of the weakly-labeled training data. However, it does *not* indicate whether that better subset actually leads to a more accurate end model. Since we could also use $\phi$ for the end model—e.g., by fine-tuning the full neural network or training a linear model on top of $\phi$—it's possible that the training step (1) will already perform the same corrections as the cut statistic, and the end model trained on the selected subset will perform no differently from the end model trained with $\beta = 1.0$. In the following section, we focus on the cut statistic and conduct large-scale empirical evaluation on the WRENCH benchmark to measure whether subset selection improves end model performance. Our empirical results suggest that subset selection and the end model training step are complementary: even when we use powerful representations for the end model, subset selection further improves performance, sometimes by a large margin.

# 4 Experiments

Having established that the cut statistic can effectively select weakly-labeled training *sub*sets that are higher-quality than the original training set, we now turn to a wider empirical study to see whether this approach actually improves the performance of end models in practice.

**Datasets and Models.** We evaluate our approach on the WRENCH benchmark [42] for weak supervision. We compare the status-quo of full coverage ($\beta = 1.0$) to $\beta$ chosen from $\{0.1, 0.2, \dots, 1.0\}$. We evaluate our approach with five different label models: Majority Vote (**MV**), the original Snorkel/Data Programming (**DP**), [30], Dawid-Skene (**DS**) [8], FlyingSquid (**FS**) [10], and **MeTaL** [29]. Following Zhang et al. [42], we use pretrained `roberta-base` and `bert-base-cased`[1] as the end model representation for text data, and hand-specified representations for tabular data. We performed all model training on NVIDIA A100 GPUs. We primarily evaluate on seven textual datasets from the WRENCH benchmark: **IMDb** (sentiment analysis), **Yelp** (sentiment analysis), **Youtube** (spam classification), **TREC** (question classification), **SemEval** (relation extraction), **ChemProt** (relation extraction), and **AGNews** (text classification). Full details for the datasets and the weak label sources are available in [42] Table 5 and reproduced here in Appendix B.1. We explore other several other datasets and data modalities in Sections 4.2-4.3.

**Cut statistic.** For the representation $\phi$, for text datasets we used the `[CLS]` token representation of a large pretrained model such as BERT or RoBERTa. For relation extraction tasks, we followed [42] and used the concatenation of the `[CLS]` token and the average contextual representation of the tokens in each entity span. In Section 4.3, for the tabular **Census** dataset we use the raw data features for $\phi$. Unless otherwise specified, we used *the same representation* for $\phi$ and for the initial end model. For example, when training `bert-base-cased` as the end model, we used `bert-base-cased` as $\phi$ for the cut statistic. We explore several alternatives to this choice in Section 4.2.

**Hyperparameter tuning.** Our subset selection approach introduces a new hyperparameter, $\beta$—the fraction of covered data to retain for training the classifier. To keep the hyperparameter tuning burden low, we first tune all other hyperparameters identically to Zhang et al. [42] holding $\beta$ fixed at 1.0. We then use the optimal hyperparameters (learning rate, batch size, weight decay, etc.) from $\beta = 1.0$ for a grid search over values of $\beta \in \{0.1, 0.2, \dots, 1.0\}$, choosing the value with the best (ground-truth) validation performance. Better results could be achieved by tuning all the hyperparameters together, but this approach limits the number of possible combinations, and it matches the setting where an existing, tuned weak supervision pipeline (with $\beta = 1.0$) is adapted to use subset selection. In all of our experiments, we used $K = 20$ nearest neighbors to compute the cut statistic and performed no tuning on this value. Appendix B contains an ablation showing that performance is not sensitive to this choice.

## 4.1 WRENCH Benchmark Performance

Table 1 compares the test performance of full coverage ($\beta = 1.0$) to the performance of the cut statistic with $\beta$ chosen according to validation performance. Standard deviations across five random initializations are shown in parentheses.

---

[1]We refer to pretrained models by their names on the HuggingFace Datasets Hub. All model weights were downloaded from the hub: https://huggingface.co/datasets

Table 1: End model test accuracy (stddev) for weak supervision with $\beta = 1$ versus weak supervision with $\beta$ selected from $\{0.1, 0.2, 0.3, \ldots, 1.0\}$ using a validation set ("+ cutstat"), shown for BERT ($B$) and RoBERTa ($RB$) end models. For these results, the cut statistic uses the same representation as the end model for $\phi$. The cut statistic broadly improves the performance of weak supervision for many (label model, dataset, end model) combinations.

| | Label model | imdb | yelp | youtube | trec | semeval | chemprot | agnews |
|---|---|---|---|---|---|---|---|---|
| **B** | Majority Vote | $78.32_{2.62}$ | $86.85_{1.42}$ | $95.12_{1.27}$ | $66.76_{1.46}$ | $85.17_{0.89}$ | $57.44_{2.01}$ | $86.59_{0.47}$ |
| | + *cutstat* | $\mathbf{81.86_{1.36}}$ | $\mathbf{89.49_{0.78}}$ | $\mathbf{95.60_{0.72}}$ | $\mathbf{71.84_{3.00}}$ | $\mathbf{92.47_{0.49}}$ | $\mathbf{57.47_{1.00}}$ | $86.26_{0.43}$ |
| | Data Programming | $75.90_{1.44}$ | $76.43_{1.29}$ | $92.48_{1.30}$ | $71.20_{1.78}$ | $71.97_{1.57}$ | $51.89_{1.60}$ | $86.01_{0.63}$ |
| | + *cutstat* | $\mathbf{79.07_{2.52}}$ | $\mathbf{88.13_{1.46}}$ | $\mathbf{93.92_{0.93}}$ | $\mathbf{76.76_{1.92}}$ | $\mathbf{91.07_{0.90}}$ | $\mathbf{55.10_{1.49}}$ | $85.89_{0.45}$ |
| | Dawid-Skene | $78.86_{1.34}$ | $88.45_{1.42}$ | $88.45_{1.42}$ | $51.04_{1.71}$ | $72.40_{1.53}$ | $44.08_{1.37}$ | $86.26_{0.56}$ |
| | + *cutstat* | $\mathbf{80.22_{1.69}}$ | $\mathbf{89.04_{1.10}}$ | $\mathbf{90.72_{1.27}}$ | $\mathbf{57.28_{2.91}}$ | $\mathbf{89.07_{1.62}}$ | $\mathbf{49.07_{1.48}}$ | $\mathbf{86.93_{0.22}}$ |
| | FlyingSquid | $77.46_{1.88}$ | $84.98_{1.44}$ | $91.52_{2.90}$ | $31.12_{2.39}$ | $31.83_{0.00}$ | $46.72_{0.96}$ | $86.10_{0.80}$ |
| | + *cutstat* | $\mathbf{80.85_{1.50}}$ | $\mathbf{88.75_{1.13}}$ | $91.04_{1.23}$ | $\mathbf{33.84_{3.17}}$ | $31.83_{0.00}$ | $\mathbf{48.65_{0.99}}$ | $85.90_{0.39}$ |
| | MeTaL | $78.97_{2.57}$ | $83.05_{1.69}$ | $93.36_{1.15}$ | $58.88_{1.22}$ | $58.17_{1.77}$ | $55.61_{1.35}$ | $86.06_{0.82}$ |
| | + *cutstat* | $\mathbf{81.49_{1.51}}$ | $\mathbf{88.41_{1.19}}$ | $92.64_{0.41}$ | $\mathbf{63.80_{2.28}}$ | $\mathbf{65.23_{0.91}}$ | $\mathbf{58.33_{0.81}}$ | $\mathbf{86.16_{0.48}}$ |
| **RB** | Majority Vote | $86.99_{0.55}$ | $88.51_{3.25}$ | $95.84_{1.18}$ | $67.60_{2.38}$ | $85.83_{1.22}$ | $57.06_{1.12}$ | $87.46_{0.53}$ |
| | + *cutstat* | $86.69_{0.75}$ | $\mathbf{95.19_{0.23}}$ | $\mathbf{96.00_{1.10}}$ | $\mathbf{72.92_{1.31}}$ | $\mathbf{92.07_{0.80}}$ | $\mathbf{59.05_{0.56}}$ | $\mathbf{88.01_{0.47}}$ |
| | Data Programming | $86.31_{1.53}$ | $88.73_{5.07}$ | $94.08_{1.48}$ | $71.40_{3.30}$ | $71.07_{1.66}$ | $52.52_{0.69}$ | $86.75_{0.24}$ |
| | + *cutstat* | $\mathbf{86.46_{1.82}}$ | $\mathbf{93.95_{0.93}}$ | $93.04_{1.30}$ | $\mathbf{76.84_{4.09}}$ | $\mathbf{86.07_{1.82}}$ | $\mathbf{56.43_{1.37}}$ | $\mathbf{87.76_{0.17}}$ |
| | Dawid-Skene | $85.50_{1.68}$ | $92.42_{1.41}$ | $92.48_{1.44}$ | $51.24_{3.50}$ | $70.83_{0.75}$ | $45.61_{2.60}$ | $87.29_{0.40}$ |
| | + *cutstat* | $\mathbf{86.14_{0.60}}$ | $\mathbf{93.81_{0.69}}$ | $\mathbf{93.84_{0.70}}$ | $\mathbf{58.48_{2.75}}$ | $\mathbf{81.67_{1.33}}$ | $\mathbf{52.93_{1.67}}$ | $\mathbf{88.35_{0.22}}$ |
| | FlyingSquid | $85.25_{1.96}$ | $92.14_{2.76}$ | $93.52_{2.11}$ | $35.40_{1.32}$ | $31.83_{0.00}$ | $47.23_{1.04}$ | $86.56_{0.55}$ |
| | + *cutstat* | $\mathbf{87.71_{0.76}}$ | $\mathbf{94.50_{0.74}}$ | $\mathbf{95.84_{0.54}}$ | $\mathbf{38.16_{0.43}}$ | $31.83_{0.00}$ | $\mathbf{50.55_{1.05}}$ | $\mathbf{87.49_{0.13}}$ |
| | MeTaL | $86.16_{1.13}$ | $88.41_{3.25}$ | $92.40_{1.19}$ | $55.44_{1.08}$ | $59.53_{1.87}$ | $56.74_{0.58}$ | $86.74_{0.60}$ |
| | + *cutstat* | $\mathbf{87.46_{0.65}}$ | $\mathbf{94.03_{0.53}}$ | $\mathbf{93.84_{1.38}}$ | $\mathbf{69.72_{2.39}}$ | $\mathbf{66.70_{0.90}}$ | $\mathbf{57.40_{0.98}}$ | $\mathbf{88.40_{0.38}}$ |

The cut statistic improves the mean performance (across runs) compared to $\beta = 1.0$ in 61/70 cases, sometimes by 10–20 accuracy points (e.g., BERT SemEval DP). Since $\beta = 1.0$ is included in the hyperparameter search over $\beta$, the only cases where the cut statistic performs worse than $\beta = 1.0$ are due to differences in the performance on the validation and test sets. The mean accuracy gain from setting $\beta < 1.0$ across all 70 trials is 3.65 points, indicating that the cut statistic is complementary to the end model training. If no validation data is available to select $\beta$, we found that $\beta = 0.6$ had the best median performance gain over all label model, dataset, and end model combinations: +1.7 accuracy

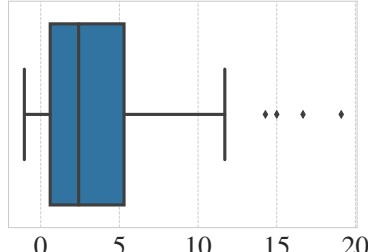

Figure 3: Test accuracy gain from setting $\beta < 1$ across all WRENCH trials.

points compared to $\beta = 1.0$. However, we show in Section 4.4 that very small validation sets are good enough to select $\beta$. The end model trains using $\phi$, but using $\phi$ to *first* select a good training set further improves performance. Figure 3 displays a box plot of the accuracy gain from using sub-selection. Appendix B.2 contains plots of the end model performance versus the coverage fraction $\beta$. In some cases, the cut statistic is competitive with COSINE [41] , which does multiple rounds of self-training on the unlabeled data. Table 7 compares the two methods, and we show in Appendix B.7 how to combine them to improve performance.

For Table 1, we used the same representation for $\phi$ and for the initial end model. These results indicate that representations from very large models such as GPT-3 or CLIP are not needed to improve end model performance. However, there is no *a priori* reason to use the same representation for $\phi$ and for the end model initialization. Using a much larger model for $\phi$ may improve the cut statistic performance without drastically slowing down training, since we only need to perform *inference* on the larger model. We examine the role of the representation choice more thoroughly in Section 4.2.

## 4.2   Choice of Representation for Cut Statistic

How important is the quality of $\phi$ (the representation used for the cut statistic) for the performance of subset selection? In this section we experiment with different choices of $\phi$. To isolate the effect of $\phi$

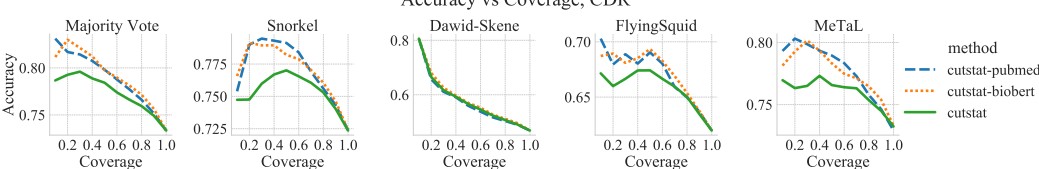

Figure 4: Domain-specific pretraining versus general-domain pretraining for $\phi$. Stock BERT (*cutstat*) compared to BioBERT (*cutstat-biobert*), and PubMedBERT (*cutstat-pubmed*), two models pretrained on biomedical text. The domain-specific models select more accurate subsets than the generic model.

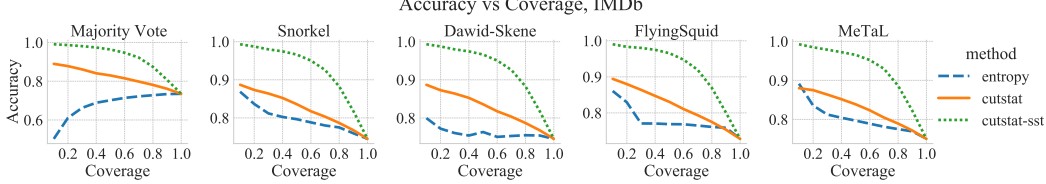

Figure 5: Comparison of IMDb training subset accuracy for the cut statistic with generic BERT (*cutstat*) and a BERT fine-tuned for sentiment analysis on the SST-2 dataset (*cutstat-sst*). The fine-tuned representation gives very high-quality training subsets when used with the cut statistic.

on performance, we use the generic BERT as the end model. Performance can improve further when using more powerful representations for the end model as well, but we use a fixed end model here to explore how the choice of $\phi$ affects final performance. Our results indicate that (i) for weak supervision tasks in a specific domain (e.g., biomedical text), models pretrained on that domain perform better than general-domain models as $\phi$ and (ii) for a specific task (e.g., sentiment analysis), models pretrained on that task, but on different data, perform very well as $\phi$. We also show in Appendix B that using a larger generic model for $\phi$ can improve performance when the end model is held fixed.

**Domain-specific pretraining can help.** The **CDR** dataset in WRENCH has biomedical abstracts as input data. Instead of using general-domain $\phi$ such as BERT and RoBERTa, does using a domain-specific version improve performance? Figure 4 shows that domain specific models *do* improve over general-domain models when used in the cut statistic. We compare `bert-base-cased` to `PubMedBERT-base-uncased-abstract-fulltext` [11] and `biobert-base-cased-v1.2` [17]. The latter two models were pretrained on biomedical text. The domain-specific models lead to higher quality training datasets for all label models except Dawid-Skene. These gains in training dataset accuracy translate to gains in end-model performance. Trained with $\hat{Y}$ from majority vote and using BioBERT as $\phi$, a general-domain BERT end model obtains test F1 score 61.14 (0.64), compared to 59.63 (0.84) using BERT for both $\phi$ and the end model. Both methods improve over 58.20 (0.55) obtained from training generic BERT with $\beta = 1.0$ (no sub-selection).

**Representations can transfer.** If a model is trained for a particular task (e..g, sentiment analysis) on one dataset, can we use it as $\phi$ to perform weakly-supervised learning on a different dataset? We compare two choices for $\phi$ on IMDb: regular `bert-base-cased`, and `bert-base-cased` fine-tuned with fully-supervised learning on the Stanford Sentiment Treebank (SST) dataset [37]. As indicated in Figure 5, the fine-tuned BERT representation selects a far higher-quality subset for training. This translates to better end model performance as well. Using majority vote with the fine-tuned BERT as $\phi$ leads to test performance of 87.22 (0.57), compared to 81.86 in Table 1. These results suggest that if we have a representation $\phi$ that's already useful for a task, we can effectively combine it with the cut statistic to improve performance on a different dataset.

### 4.3 Other Data Modalities

Our experiments so far have used text data, where large pretrained models like BERT and RoBERTa are natural choices for $\phi$. Here we briefly study the cut statistic on *tabular* and *image* data. The **Census** dataset in WRENCH consists of tabular data where the goal is to classify whether a person's income is greater than \$50k from a set of 13 features. We also use these hand-crafted features for $\phi$ and train a linear model on top of the features for the end model. The **Basketball** dataset is a set of still images obtained from videos, and the goal is to classify whether basketball is being

Table 2: Test F1 of a weakly-supervised linear model (*LR*) on the **Census** dataset, which consists of 13 hand-created features. Even though the representation does not come from a large, pretrained neural network, the cut statistic improves the performance of weak supervision for every label model. Test F1 of a 1-hidden-layer network (*MLP*) on CLIP representations of the **Basketball** dataset, where the results are noisy, but the cut statistic improves the mean performance of every label model.

| | Dataset | **Majority Vote** | **Data Programming** | **Dawid-Skene** | **FlyingSquid** | **MeTaL** |
|---|---|---|---|---|---|---|
| LR | Census | $50.71_{2.18}$ | $21.67_{17.32}$ | $49.90_{0.61}$ | $38.23_{3.96}$ | $51.41_{1.45}$ |
| | + *cutstat* | $\mathbf{57.98}_{0.68}$ | $\mathbf{28.04}_{17.38}$ | $\mathbf{58.49}_{0.23}$ | $\mathbf{40.53}_{2.26}$ | $\mathbf{54.99}_{1.54}$ |
| MLP | Basketball | $52.29_{6.62}$ | $52.14_{4.80}$ | $22.59_{9.83}$ | $54.04_{12.15}$ | $32.99_{12.98}$ |
| | + *cutstat* | $\mathbf{55.82}_{3.98}$ | $\mathbf{54.59}_{14.80}$ | $\mathbf{43.77}_{12.47}$ | $\mathbf{56.60}_{6.13}$ | $\mathbf{47.97}_{8.89}$ |

Table 3: Comparison between using the full validation set to choose $\beta$ and the model checkpoint versus using a randomly selected validation subset of 100 examples. These results use the majority vote (MV) label model. Standard deviation is reported over five random seeds used to select the validation set (not to be confused with Table 1, where standard deviation is reported over random seeds controlling the deep model initialization). Most of the drop in performance is due to the noisier checkpoint selection when using the small validation set. I.e., the difference between $\beta =$ best and $\beta = 1.0$ is similar for the full validation and random validation cases.

| | Val. size | $\beta$ | **imdb** | **yelp** | **youtube** | **trec** | **semeval** | **chemprot** | **agnews** |
|---|---|---|---|---|---|---|---|---|---|
| BERT | full | 1.0 | 78.32 | 86.85 | 95.12 | 66.76 | 85.17 | 57.44 | 86.59 |
| | full | best | 81.86 | 89.49 | 95.60 | 71.84 | 92.47 | 57.47 | 86.26 |
| | 100 | 1.0 | $79.17_{2.80}$ | $84.88_{1.97}$ | $94.50_{0.32}$ | $65.33_{1.84}$ | $85.62_{0.64}$ | $54.99_{1.63}$ | $84.77_{1.11}$ |
| | 100 | best | $79.75_{2.18}$ | $87.96_{1.00}$ | $94.40_{0.63}$ | $74.50_{1.83}$ | $92.40_{1.81}$ | $54.40_{2.35}$ | $84.96_{1.20}$ |
| RoBERTa | full | 1.0 | 86.99 | 88.51 | 95.84 | 67.60 | 85.83 | 57.06 | 87.46 |
| | full | best | 86.69 | 95.19 | 96.00 | 72.92 | 92.07 | 59.05 | 88.01 |
| | 100 | 1.0 | $85.74_{1.11}$ | $89.32_{1.49}$ | $95.24_{1.11}$ | $66.40_{1.56}$ | $84.38_{1.18}$ | $56.71_{0.55}$ | $85.79_{0.75}$ |
| | 100 | best | $85.24_{0.78}$ | $93.82_{0.48}$ | $96.06_{0.70}$ | $75.15_{2.89}$ | $91.24_{0.76}$ | $56.52_{0.44}$ | $87.20_{0.31}$ |

played in the image using the output of an off-the-shelf object detector in the $\Lambda_k$'s. We used CLIP [27] representations of the video frames and trained a 1-hidden-layer neural network using the hyperparameter tuning space from [42]. Table 2 shows the results for these datasets. The cut statistic improves the end model performance for every label model even with the small, hand-crafted representation, and also improves for the **Basketball** data.

## 4.4 Using a Smaller Validation Set

Many datasets used to evaluate weak supervision methods actually come with large labeled validation sets. For example, the average validation set size of the WRENCH datasets from Table 1 is over 2,500 examples. However, assuming access to a large amount of labeled validation data partially defeats the purpose of weak supervision. In this section, we show that the coverage parameter $\beta$ for the cut statistic can be selected using a much smaller validation set without compromising the performance gain over $\beta = 1.0$. We compare choosing the best model checkpoint and picking the best coverage fraction $\beta$ using (i) the *full* validation set and (ii) a randomly-sampled validation set of 100 examples. Table 3 shows the results for the majority vote label model. The full validation numbers come from Table 1. The difference between selecting data with the validation-optimal $\beta$ and using $\beta = 1.0$ is broadly similar between the full validation and small validation cases. This suggests that most of the drop in performance from full validation to small validation is due to the noisier choice of the best model checkpoint, not due to choosing a suboptimal $\beta$.

## 4.5 Discussion

**Why not *correct* pseudolabels with nearest neighbor?** Consider an example $x_i$ whose weak label $\hat{Y}(x_i)$ disagrees with the weak label $\hat{Y}(x_j)$ of most neighbors $j \in N(i)$. This example would get thrown out by the cut statistic selection. Instead of throwing such data points out, we could try to *re-label* them with the majority weak label from the neighbors. However, throwing data out is a more conservative (and hence possibly more robust) approach. For example, if the weak labels are

mostly *wrong* on hard examples close to the true unknown decision boundary, relabeling makes the training set worse, whereas the cut statistic ignores these points. Appendix B.3 contains an empirical comparison between subset selection and relabeling. For the representations studied in this work, relabeling largely fails to improve training set quality and end model performance.

**Why does sub-selection work?** As suggested above, subset selection can change the distribution of data points in the training set shown to the end model. For example, it may only select "easy" examples. However, this is already a problem in today's weak supervision methods: the full weakly-labeled training set $\mathcal{T}$ is already biased. For example, many labeling functions are keyword-based, such as those in Section 2 ("good"→positive sentiment, "bad"→negative). In these examples, $\mathcal{T}$ itself is a biased subset of the input distribution (only sentences that contain "good" or "bad", versus all sentences). Theoretical understanding for why weak supervision methods perform well on the uncovered set $\mathcal{X} \backslash \{x : \mathbf{\Lambda}(x) \neq \varnothing\}$ is currently lacking, and existing generalization bounds for the end model do not capture this phenomenon. In the following section we present a special (but practically-motivated) case where this bias can be avoided. In this case, we prove a closed form for the coverage-precision tradeoff of selection methods, giving subset selection some theoretical motivation.

## 5  Theoretical Results: Why Does Subset Selection Work?

We begin by presenting a theoretical setup motivated by the CheXpert [13] and MIMIC-CXR [14] datasets, where the weak labels are derived from radiology notes and the goal is to learn an end model for classifying X-ray images. Suppose for this section that we have two (possibly related) *views* $\psi_0(X)$, $\psi_1(X)$ of the data $X$, i.e., $\psi_0 : \mathcal{X} \to \Psi_0$, $\psi_1 : \mathcal{X} \to \Psi_1$. We use $\psi$ here to distinguish from $\phi$, the representation used to compute nearest neighbors for the cut statistic. For example, if the input space $\mathcal{X}$ is multi-modal, and each $x_i = (x_i^{(0)}, x_i^{(1)})$, then we can set $\psi_0$ and $\psi_1$ to project onto the individual modes (e.g., $\phi_0(X)$ the clinical note and $\phi_1(X)$ the X-ray). We will assume that the labeling functions $\Lambda_k(x_i)$ only depend on $\psi_0(x_i)$, and that the end model $f$ only depends on $\psi_1(x_i)$. In the multi-modal example, this means the labeling functions are defined on one view, and the end model is trained on the other view. To prove a closed form for the precision/coverage tradeoff, we make the following strong assumption relating the two views $\psi_0$ and $\psi_1$:

**Assumption 1** (Conditional independence). *The random variables $\psi_0(X)$, $\psi_1(X)$ are conditionally independent given the true (unobserved) label $Y$. That is, for any sets $A \subset \Psi_0$, $B \subset \Psi_1$,*
$$\mathbb{P}_{X,Y}[\psi_0(X) \in A, \psi_1(X) \in B | Y] = \mathbb{P}_{X,Y}[\psi_0(X) \in A | Y] \mathbb{P}_{X,Y}[\psi_1(X) \in B | Y].$$

Note since every $\Lambda_k$ only depends on $\psi_0(X)$, the pseudolabel $\hat{Y}$ only depends on $\psi_0(X)$. Hence $\hat{Y}(X) = \hat{Y}(\psi_0(X))$, and likewise for an end model $f$, $f(X) = f(\psi_1(X))$. Assumption 1 implies:
$$\mathbb{P}_{X,Y}[\mathbb{I}[\hat{Y}(X) \neq Y], \psi_1(X) \in B | Y] = \mathbb{P}_{X,Y}[\mathbb{I}[\hat{Y}(X) \neq Y] | Y] \mathbb{P}_{X,Y}[\psi_1(X) \in B | Y]$$
for every $B \subset \Psi_1$. In this special case, the end model training reduces to learning with class-conditional noise (CCN), since the errors $\mathbb{I}[\hat{Y}(X) \neq Y]$ are conditionally independent of the representation $\psi_1(X)$ being used for the end model. This assumption is most natural for the case of multi-modal data and $\psi_0$, $\psi_1$ that project onto each mode, but it may also roughly apply when the representation being used for the end model (such as a BERT representation) is "suitably orthogonal" to the input $X$. While very restrictive, this assumption allows us to make the coverage-precision tradeoff precise.

**Theorem 1.** *Suppose Assumption 1 holds, and that $\mathcal{Y} = \{0, 1\}$. Define the* balanced error *of a classifier $f$ on labels $Y$ as: $err_{bal}(f, Y) = \frac{1}{2}(\mathbb{P}[f(X) = 0 | Y = 1] + \mathbb{P}[f(X) = 1 | Y = 0])$. We write $f(X)$ instead of $f(\psi_1(X))$ for convenience. Let $\hat{Y} : \Psi_0(X) \to \{0, 1\}$ be an arbitrary label model. Define $\alpha = \mathbb{P}[Y = 0 | \hat{Y} = 1]$ and $\gamma = \mathbb{P}[Y = 1 | \hat{Y} = 0]$ and suppose $\alpha + \gamma < 1$, $\mathbb{P}[\hat{Y} = y] > 0$ for $y \in \{0, 1\}$. These parameters measure the amount of noise in $\hat{Y}$. Define $f^* := \inf_{f \in \mathcal{F}} err_{bal}(f, Y)$. Let $\hat{f}$ be the classifier obtained by minimizing the empirical balanced accuracy on $\{(x_i, \hat{Y}(x_i))\}_{i=1}^n$. Then the following holds w.p. $1 - \delta$ over the sampling of the data:*
$$err_{bal}(\hat{f}, Y) - err_{bal}(f^*, Y) \leq \tilde{\mathcal{O}}\left(\frac{1}{1 - \alpha - \gamma}\sqrt{\frac{VC(\mathcal{F}) + \log\frac{1}{\delta}}{n\mathbb{P}[\hat{Y} \neq \varnothing]\min_y \mathbb{P}[\hat{Y} = y | \hat{Y} \neq \varnothing]}}\right),$$

*where $\tilde{\mathcal{O}}$ hides log factors in $m$ and $VC(\mathcal{F})$.*

*Proof.* For space, we defer the proof and and bibliographic commentary to Appendix A. $\qquad\square$

This bound formalizes the tradeoff between the *precision* of the weak labels, measured by $\alpha$ and $\gamma$, and the *coverage*, measured by $n\mathbb{P}[Y \neq \varnothing]$, which for large enough samples is very close to the size of the *covered* training set $\mathcal{T} = \{(x_i, \hat{Y}(x_i)) : \hat{Y}(x_i) \neq \varnothing\}$. Suppose we have a label model $\hat{Y}$ and an alternative label model $\hat{Y}'$ that abstains more often than $\hat{Y}$ (so $\mathbb{P}[Y' \neq \varnothing] < \mathbb{P}[\hat{Y} \neq \varnothing]$) but also has smaller values of $\alpha$ and $\gamma$. Then according to the bound, an end model trained with $\hat{Y}'$ can have better performance, and the empirical results in Section 4 confirm this trend.

This bound is useful for comparing two fixed label models $\hat{Y}, \hat{Y}'$ with different abstention rates and $(\alpha, \gamma)$ values. However, we have been concerned in this paper with selecting a subset $\mathcal{T}'$ of $\mathcal{T}$ based on a single label model $\hat{Y}$, and training using $\mathcal{T}'$. We can represent this subset selection with a new set of pseudolabels $\{\tilde{Y}(x_i) : x_i \in \mathcal{T}\}$ that abstains more than $\hat{Y}(x_i)$—i.e., points not chosen for $\mathcal{T}'$ get $\tilde{Y}(x_i) = \varnothing$. However, selection for $\mathcal{T}'$ depends on sample-level statistics, so the $\tilde{Y}$ values are not i.i.d., which complicates the generalization bound. We show in Appendix A that this can be remedied by a sample-splitting procedure: we use half of $\mathcal{T}$ to define a refined label model $\tilde{Y} : \Psi_0 \to \mathcal{Y} \cup \{\varnothing\}$, and then use the other half of $\mathcal{T}$ as the initial training set. This allows us to effectively reduce to the case of two fixed label models $\hat{Y}, \tilde{Y}$ and apply Theorem 1. We include the simpler $\hat{Y}$ versus $\hat{Y}'$ bound here because it captures the essential tradeoff without the technical difficulties.

# 6   Limitations, Societal Impact, and Conclusion

Surprisingly, using *less* data can greatly improve the performance of weak supervision pipelines when that data is carefully selected. By exploring the tradeoff between weak label precision and coverage, subset selection allows us to select a higher-quality training set without compromising generalization performance to the population pseudolabeling function. This improves the accuracy of the end model on the true labels. In Section 5, we showed that this tradeoff can be formalized in the special setting of conditional independence. By combining the cut statistic with good data representations, we developed a technique that improves performance for five different label models, over ten datasets, and three data modalities. Additionally, the hyperparameter tuning burden is low. We introduced one new hyperparameter $\beta$ (the coverage fraction) and showed that all other hyperparameters can be re-used from the full-coverage $\beta = 1.0$ case, so existing tuned weak supervision pipelines can be easily adapted to use this technique.

However, this approach is not without limitations. The cut statistic requires a good representation $\phi$ of the input data to work well. Such a representation may not be available. However, for image or text data, pretrained representations provide natural choices for $\phi$. Our results on the Census dataset in Section 4.3 indicate that using hand-crafted features as $\phi$ can also work well. Finally, as discussed at the end of Section 4.5, subset selection can further bias the input distribution (except in special cases like the one in Section 5). However, this is already an issue with current weak supervision methods. Most methods only train on the covered data $\mathcal{T}$. Labeling functions are typically deterministic functions of the input example, (such as functions based on the presence of certain tokens) and so the support of the full training set $\mathcal{T}$ is a strict subset of the support of the true input distribution, and $\mathcal{T}$ may additionally have a skewed distribution over its support. This underscores the need for (i) the use of a ground-truth validation set to ensure that the end model is an accurate predictor on the full distribution (ii) in high stakes settings, sub-group analyses such as those performed by [35], to ensure that the pseudolabels have not introduced bias against protected subgroups and (iii) the need for further theoretical understanding on why weakly supervised end models are able to perform well on the uncovered set $\{x : \Lambda(x) = \varnothing\}$.

## Acknowledgments and Disclosure of Funding

This work was supported by NSF AitF awards CCF-1637585 and CCF-1723344. Thanks to Hussein Mozannar for helpful conversations on Section 5 and the pointer to Woodworth et al. [39]. Thanks to Dr. Steven Horng for generously donating GPU-time on the BIDMC computing cluster [12] and to NVIDIA for their donation of GPUs also used in this work.

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
