# A Proofs from Section 5

To ease notation in this section, we consider the multimodal setting $\mathcal{X} = \mathcal{X}^{(0)} \times \mathcal{X}^{(1)}$. The extension to arbitrary $\phi_0(X)$, $\phi_1(X)$ is straightforward. First, we fix an arbitrary label model $\hat{Y}$ and, we assume for this part that $\hat{Y}$ can be written as a function mapping single examples to pseudolabels: $\hat{Y} : \mathcal{X}^{(0)} \to \mathcal{Y} \cup \{\varnothing\}$. We first prove Theorem 1 for this case. Second, we discuss the extention of this theorem to the case where the pseudolabels $\tilde{Y}$ come from some label model $\hat{Y}$ plus a subset selector such as the cut statistic. This complicates the situation because the post-subselection pseudolabels $\{\tilde{Y}(x_i)\}$ (i.e., the output of the cut statistic on the training set) cannot be written as i.i.d. samples of a "refined label model" $\tilde{Y} : \mathcal{X}^{(0)} \to \mathcal{Y} \cup \{\varnothing\}$. We show how to use sample splitting to suitably define the population-level function $\tilde{Y} : \mathcal{X}^{(0)} \to \mathcal{Y} \cup \{\varnothing\}$, which allows us to directly apply Theorem 1.

## A.1 Proof of Theorem 1.

Suppose that $\mathcal{Y} = \{0, 1\}$. Recall the conditional independence assumption:

**Assumption** (Conditional independence). *The random variables $X^{(0)}$, $X^{(1)}$ are conditionally independent given the true (unobserved) label $Y$. That is, for any $A \subset \mathcal{X}^{(0)}$, $B \subset \mathcal{X}^{(1)}$,*

$$\mathbb{P}_{X,Y}[X^{(0)} \in A, X^{(1)} \in B|Y] = \mathbb{P}_{X,Y}[X^{(0)} \in A|Y]\mathbb{P}_{X,Y}[X^{(1)} \in B|Y].$$

Let $\hat{Y} : \mathcal{X}^{(0)} \to \mathcal{Y} \cup \{\varnothing\}$ be an arbitrary label model, and define:

$$\alpha = \mathbb{P}_{X,Y}[Y = 0|\hat{Y}(X^{(0)}) = 1]$$
$$\gamma = \mathbb{P}_{X,Y}[Y = 1|\hat{Y}(X^{(0)}) = 0]$$

These parameters measure the amount of noise in $\hat{Y}$. We assume throughout that $\mathbb{P}[\hat{Y} = y] > 0$ for $y \in \{0, 1\}$ and that $\alpha + \gamma < 1$. Note that this implies $\mathbb{P}[\hat{Y} \neq \varnothing|Y = y] > 0$ for all $y$, since otherwise either $\alpha$ or $\gamma$ is 1. So there are pseudolabeled examples from both conditional distributions $\mathbb{P}[X|Y = y]$.

**Theorem.** *Suppose we observe a weakly labeled training set $\bar{\mathcal{T}} := \{x_i^{(0)}, x_i^{(1)}, \hat{Y}(x_i^{(0)})\}_{i=1}^n$, where the $x_i$'s are drawn i.i.d. from the marginal distribution $\mathbb{P}_X$. This differs from $\mathcal{T}$ in the main text because we include both views $x^{(0)}, x^{(1)}$ and also include the points where $\hat{Y}(x_i^{(0)}) = \varnothing$.*

*Let $\mathcal{F}$ be a hypothesis class consisting of functions $f : \mathcal{X}^{(1)} \to \mathcal{Y}$. Suppose Assumption 1 holds, and that $\mathbb{P}[\hat{Y} = y] > 0$ for $y \in \{0, 1\}$ and that $\alpha + \gamma < 1$. Define the* balanced error *of a classifier $f$ on labels $Z \in \{0, 1\}$ as:*

$$err_{bal}(f, Z) = \frac{1}{2}(\mathbb{P}[f(X) = 0|Z = 1] + \mathbb{P}[f(X) = 1|Z = 0]).$$

*We will consider both $Z = Y$ and $Z = \hat{Y}$. Let $f \in \mathcal{F}$ be an arbitrary classifier. Then $f$'s true balanced error $err_{bal}(f, Y)$ and $f$'s pseudolabel balanced error $err_{bal}(f, \hat{Y})$ satisfy:*

$$err_{bal}(f, Y) = \frac{1}{1 - \alpha - \gamma}\left[err_{bal}(f, \hat{Y}) - \frac{\alpha + \gamma}{2}\right].$$

*Now let $f^* := \inf_{f \in \mathcal{F}} err_{bal}(f, Y)$ be the classifier with optimal balanced accuracy on the true labels. Suppose that $\hat{f}$ is the classifier obtained by minimizing the empirical balanced accuracy on the dataset $\mathcal{T} = \{(x_i^{(1)}, \hat{Y}(x_i^{(0)}))\}$:*

$$\hat{f} := \underset{f \in \mathcal{F}}{argmin}\ \widehat{err}_{bal}(f, \hat{Y}) = \frac{1}{2}\left(\frac{\sum_{i=1}^n \mathbb{1}_{f(x_i^{(1)})=0}\mathbb{1}_{\hat{Y}(x_i^{(0)})=1}}{\sum_{i=1}^n \mathbb{1}_{\hat{Y}(x_i^{(0)})=1}} + \frac{\sum_{i=1}^n \mathbb{1}_{f(x_i^{(1)})=1}\mathbb{1}_{\hat{Y}(x_i^{(0)})=0}}{\sum_{i=1}^n \mathbb{1}_{\hat{Y}(x_i^{(0)})=0}}\right).$$

*Note that the training points $\{x_i : \hat{Y}(x_i^{(0)}) = \varnothing\}$ do not feature at all in the above expression, so we can safely discard them for the training step. Then for any $\delta > 0$ the following holds with probability*

*at least $1 - \delta$ over the sampling of $\bar{\mathcal{T}}$:*

$$err_{bal}(f, Y) - err_{bal}(f^*, Y) \leqslant \tilde{\mathcal{O}} \left( \frac{1}{1 - \alpha - \gamma} \sqrt{\frac{VC(\mathcal{F}) + \log \frac{1}{\delta}}{n\mathbb{P}[\hat{Y} \neq \varnothing] \min_y \mathbb{P}[\hat{Y} = y | \hat{Y} \neq \varnothing]}} \right),$$

*where $\tilde{\mathcal{O}}$ hides log factors in $m$ and $VC(\mathcal{F})$.*

*Proof.* Lemma 1 proves that for any $f \in \mathcal{F}$,

$$\text{err}_{bal}(f, Y) = \frac{1}{1 - \alpha - \gamma} \left[ \text{err}_{bal}(f, \hat{Y}) - \frac{\alpha + \gamma}{2} \right].$$

Subtracting $\text{err}_{bal}(f^*, \hat{Y})$ from both sides:

$$\text{err}_{bal}(\hat{f}, Y) - \text{err}_{bal}(f^*, Y) = \frac{1}{1 - \alpha - \gamma} \left[ \text{err}_{bal}(\hat{f}, \hat{Y}) - \text{err}_{bal}(f^*, \hat{Y}) \right].$$

Let $\hat{f}^*$ be the classifier in $\mathcal{F}$ with optimal population-level balanced error on $\hat{Y}$:

$$\hat{f}^* := \underset{f \in \mathcal{F}}{\operatorname{argmin}} \operatorname{err}_{bal}(f, \hat{Y}).$$

Then

$$\text{err}_{bal}(\hat{f}, Y) - \text{err}_{bal}(f^*, Y) \leqslant \frac{1}{1 - \alpha - \gamma} \left[ \text{err}_{bal}(\hat{f}, \hat{Y}) - \text{err}_{bal}(\hat{f}^*, \hat{Y}) \right]. \tag{2}$$

Now we need to control $\text{err}_{bal}(\hat{f}, \hat{Y}) - \text{err}_{bal}(\hat{f}^*, \hat{Y})$, the excess risk of $\hat{f}$ on the weak labels $\hat{Y}$. From $\bar{T}$, we can form a sample of $n$ i.i.d. points $\{(x_i^{(1)}, \hat{y}_i)\}_{i=1}^n$ from the joint distribution: $\mathbb{P}_X[X^{(1)}, \hat{Y}(X^{(0)})]$.

Theorem 2 implies that for any $\delta > 0$, with probability at least $1 - \delta$ over the sampling of $\{(x_i^{(1)}, \hat{y}_i)\}_{i=1}^n$, we have the following deviation bound:

$$\sup_{f \in \mathcal{F}} \mathbb{P}[|\widehat{\text{err}}_{bal}(f, \hat{Y}) - \text{err}_{bal}(f, \hat{Y})|] \leqslant \tilde{\mathcal{O}} \left( \sqrt{\frac{VC(\mathcal{F}) + \log \frac{1}{\delta}}{n\mathbb{P}[\hat{Y} \neq \varnothing] \min_y \mathbb{P}[\hat{Y} = y | \hat{Y} \neq \varnothing]}} \right).$$

We prove Theorem 2 in the self-contained Section A.3. We can easily turn this uniform convergence result into an excess risk bound for $\hat{f}$ with a standard sequence of inequalities. We drop the subscript and remove $\hat{Y}$ from the error arguments for convenience, so $\text{err}(\cdot)$ in the following refers to $\text{err}_{bal}(\cdot, \hat{Y})$:

$$\begin{aligned} \text{err}(\hat{f}) - \text{err}(\hat{f}^*) &= \widehat{\text{err}}(\hat{f}) - \widehat{\text{err}}(\hat{f}^*) + \text{err}(f) - \widehat{\text{err}}(f) + \widehat{\text{err}}(\hat{f}^*) - \text{err}(\hat{f}^*) \\ &\leqslant \text{err}(f) - \widehat{\text{err}}(f) + \widehat{\text{err}}(\hat{f}^*) - \text{err}(\hat{f}^*) \\ &\leqslant |\text{err}(f) - \widehat{\text{err}}(f)| + |\widehat{\text{err}}(\hat{f}^*) - \text{err}(\hat{f}^*)| \\ &\leqslant_{\text{w.p. } 1 - \delta} \tilde{\mathcal{O}} \left( \sqrt{\frac{VC(\mathcal{F}) + \log \frac{1}{\delta}}{n\mathbb{P}[\hat{Y} \neq \varnothing] \min_y \mathbb{P}[\hat{Y} = y | \hat{Y} \neq \varnothing]}} \right). \end{aligned}$$

The first inequality used that $\widehat{\text{err}}(\hat{f}) - \widehat{\text{err}}(\hat{f}^*) \leqslant 0$ since $\hat{f}$ is the empirical minimizer, and the last inequality applied the deviation bound to each term. Hence we have shown that with probability at least $1 - \delta$,

$$\text{err}_{bal}(\hat{f}, \hat{Y}) - \text{err}_{bal}(\hat{f}^*, \hat{Y}) \leqslant \tilde{\mathcal{O}} \left( \sqrt{\frac{VC(\mathcal{F}) + \log \frac{1}{\delta}}{n\mathbb{P}[\hat{Y} \neq \varnothing] \min_y \mathbb{P}[\hat{Y} = y | \hat{Y} \neq \varnothing]}} \right).$$

Plugging this in to (2) completes the proof of Theorem 1. □

**Lemma 1** ([33], [22]). *Suppose Assumption 1 holds and that $\alpha + \gamma < 1$, $\mathbb{P}[\hat{Y} = y] > 0$ for $y \in \{0, 1\}$. Then for any $f \in \mathcal{F}$, the balanced errors on $\hat{Y}$ and $Y$ satisfy the following relationship:*

$$err_{bal}(f, Y) = \frac{1}{1 - \alpha - \gamma} \left[ err_{bal}(f, \hat{Y}) - \frac{\alpha + \gamma}{2} \right].$$

*Proof.* The formula relating $err_{bal}(f, \hat{Y})$ and $err_{bal}(f, Y)$ is due to Scott et al. [33], Menon et al. [22]. We reprove it here to show that $\mathbb{P}[\hat{Y} = \varnothing] > 0$ does not affect the result, since those works consider $\hat{Y}(x_i^{(0)}) \in \{0, 1\}$. Define:

$$\widehat{\text{FNR}}(f) = \mathbb{P}[f(X^{(1)}) = 0 | \hat{Y}(X^{(0)}) = 1]$$
$$\widehat{\text{FPR}}(f) = \mathbb{P}[f(X^{(1)}) = 1 | \hat{Y}(X^{(0)}) = 0]$$
$$\text{FNR}(f) = \mathbb{P}[f(X^{(1)}) = 0 | Y = 1]$$
$$\text{FPR}(f) = \mathbb{P}[f(X^{(1)}) = 1 | Y = 0].$$

Observe that:

$$\widehat{\text{FNR}}(f) = \mathbb{P}[f(X^{(1)}) = 0 | \hat{Y}(X^{(0)}) = 1] = \sum_y \mathbb{P}[f(X^{(1)}) = 0, Y = y | \hat{Y}(X^{(0)}) = 1]$$

$$= \sum_y \frac{\mathbb{P}[f(X^{(1)}) = 0, \hat{Y}(X^{(0)}) = 1 | Y = y] \mathbb{P}[Y = y]}{\mathbb{P}[\hat{Y}(X^{(0)}) = 1]}$$

$$= \sum_y \frac{\mathbb{P}[f(X^{(1)}) = 0 | Y = y] \mathbb{P}[\hat{Y}(X^{(0)}) = 1 | Y = y] \mathbb{P}[Y = y]}{\mathbb{P}[\hat{Y}(X^{(0)}) = 1]}$$

$$= \sum_y \mathbb{P}[f(X^{(1)}) = 0 | Y = y] \mathbb{P}[Y = y | \hat{Y}(X^{(0)}) = 1]$$

$$= \mathbb{P}[f = 0 | Y = 0] \mathbb{P}[Y = 0 | \hat{Y} = 1] + \mathbb{P}[f = 0 | Y = 1] \mathbb{P}[Y = 1 | \hat{Y} = 1]$$

$$= (1 - \text{FPR}(f))\alpha + \text{FNR}(f)(1 - \alpha).$$

Similarly,

$$\widehat{\text{FPR}}(f) = (1 - \text{FNR}(f))\gamma + \text{FPR}(f)(1 - \gamma).$$

Collecting these equalities gives:

$$\begin{bmatrix} (1 - \gamma) & -\gamma \\ -\alpha & (1 - \alpha) \end{bmatrix} \begin{bmatrix} \text{FPR}(f) \\ \text{FNR}(f) \end{bmatrix} + \begin{bmatrix} \gamma \\ \alpha \end{bmatrix} = \begin{bmatrix} \widehat{\text{FPR}}(f) \\ \widehat{\text{FNR}}(f) \end{bmatrix}$$

The coefficient matrix is invertible since we assumed $\alpha + \gamma < 1$. Multiplying both sides by its inverse gives:

$$\text{FPR}(f) = \frac{1}{1 - \alpha - \gamma} \left( (1 - \alpha)\widehat{\text{FPR}}(f) + \gamma \widehat{\text{FNR}}(f) - \gamma \right)$$

$$\text{FNR}(f) = \frac{1}{1 - \alpha - \gamma} \left( \alpha \widehat{\text{FPR}}(f) + (1 - \gamma)\widehat{\text{FNR}}(f) - \alpha \right)$$

Finally, plugging these in to $err_{bal}(f, Y) = \frac{1}{2}(\text{FPR}(f) + \text{FNR}(f))$ gives the first result. $\square$

## A.2 Dealing with subset selection

Suppose $\hat{Y}$ is some fixed label model (such as majority vote). Let $\bar{\mathcal{T}} = \{(x_i^{(0)}, x_i^{(1)}, \hat{Y}(x_i^{(0)}))\}_{i=1}^n$ be the full weakly-labeled sample, including points where $\hat{Y} = \varnothing$. We have assumed the $x_i$'s are drawn i.i.d. from some distribution $\mathbb{P}_X$ over $\mathcal{X}$ satisfying:

$$\mathbb{P}[X^{(0)} \in A, X^{(1)} \in B | Y] = \mathbb{P}[X^{(0)} \in A | Y] \mathbb{P}[X^{(1)} \in B | Y]$$

for any $A \subset \mathcal{X}^{(0)}$, $B \subset \mathcal{X}^{(1)}$. Let $\mathcal{S}$ be a function that maps sets $\{(x_i^{(0)}, x_i^{(1)}, \hat{Y}(x_i^{(0)}))\}_{i=1}^n$ to $\{0,1\}^n$, where $\mathcal{S}(\bar{\mathcal{T}})_i$ indicates whether to include example $i$ in the subset. We assume that $\mathcal{S}$ only uses the information $\{(x_i^{(0)}, \hat{Y}(x_i^{(0)}))\}$, i.e., that membership in the subset does not depend directly on $x_i^{(1)}$. We can also consider $\mathcal{S}$ as defining a new set of pseudolabels $\tilde{Y}(x_i^{(0)})$ for $\bar{\mathcal{T}}$. Define these new pseudolabels $\tilde{Y}(x_i^{(0)}) \in \mathcal{Y} \cup \{\varnothing\}$ as:

$$\tilde{Y}(x_i^{(0)}) = \begin{cases} \hat{Y}(x_i^{(0)}) & \mathcal{S}(\mathcal{T})_i = 1 \\ \varnothing & \mathcal{S}(\mathcal{T})_i = 0. \end{cases}$$

We then use the refined set $\mathcal{T}' = \{(x_i^{(1)}, \tilde{Y}(x_i^{(0)}))\}$ to train the end model (recall that the end model only uses $x^{(1)}$, and that points where $\tilde{Y}(x_i^{(0)}) = \varnothing$ can be safely ignored during the training process, since they do not appear in the loss function.

Importantly, the $\mathcal{S}$ we have studied in this work operate at the set level, and so *cannot* be written as

$$\mathcal{S}(\bar{\mathcal{T}}) = \{\tilde{\mathcal{S}}(x_1^{(0)}), \tilde{\mathcal{S}}(x_2^{(0)}), \ldots, \tilde{\mathcal{S}}(x_m^{(0)})\} \in \{0,1\}^n$$

for an *example*-level selector function $\tilde{\mathcal{S}} : \mathcal{X} \times (\mathcal{Y} \cup \{\varnothing\}) \rightarrow \{0,1\}$. For example, both the cut statistic and entropy scoring use a *percentile-based* ranking of examples, and choose the top $\beta$ fraction of examples in $\bar{\mathcal{T}}$ to have $\mathcal{S}(\bar{\mathcal{T}})_i = 1$. The value of the threshold for inclusion in the subset $\mathcal{T}'$ thus depends on the entire sample $\mathcal{T}$ and not just on a single example.

Ultimately, we would like to obtain a generalization bound for an end model trained with the refined pseudolabels $\tilde{Y}$. However, the set-level form of $\mathcal{S}$ presents an issue, because it is unclear how to even define the population-level quantities considered in Theorem 1, such as $\alpha = \mathbb{P}_{X,Y}[Y = 0 | \tilde{Y} = 1]$. How do we define $\tilde{Y}(x)$ for $x$ that do not appear in the sample $\bar{\mathcal{T}}$? Additionally, the samples $\{(x_i^{(1)}, \tilde{Y}(x_i^{(0)}))\}_{i=1}^n$ are not i.i.d., which means we can't directly apply Theorem 1. To see this, consider the case where $\beta = \frac{1}{m} + \epsilon$, i.e., the selection percentile is set so that only one example is chosen to have $\tilde{Y}(x_i^{(0)}) \neq \varnothing$. Then observing a sample $(x_i^{(1)}, \tilde{Y}(x_i^{(0)}))$ with $\tilde{Y}(x_i^{(0)}) \neq \varnothing$ implies that $\tilde{Y}(x_j^{(0)}) = \varnothing$ for all $j \neq i$. To resolve these issues, we use a straightforward sample-splitting scheme.

Suppose we partition $\bar{\mathcal{T}}$ into two halves, $\bar{\mathcal{T}} = (\bar{\mathcal{T}}_0, \bar{\mathcal{T}}_1)$. We can use $\bar{\mathcal{T}}_0$ to produce an example-level selector $\mathcal{S}$, i.e., a selector such that for any $U \subset (\mathcal{X} \times \mathcal{Y})^n$, $U = \{(x_i, \hat{Y}(x_i))\}_{i=1}^n$:

$$\mathcal{S}(U) = \{\tilde{\mathcal{S}}(x_1^{(0)}), \tilde{\mathcal{S}}(x_2^{(0)}), \ldots, \tilde{\mathcal{S}}(x_n^{(0)})\} \in \{0,1\}^m.$$

for some $\tilde{\mathcal{S}} : \mathcal{X}^{(0)} \rightarrow \{0,1\}$.

For example, for the cut statistic, to compute $\tilde{\mathcal{S}}(x^{(0)})$ for an arbitrary $x$, we first compute $\hat{Y}(x^{(0)})$, and then compute the $K$-nearest neighbors of $x$ among the non-abstaining points in $\bar{\mathcal{T}}_0$. I.e., we insert $x$ into the nearest-neighbor graph over $\bar{\mathcal{T}}_0$ with the pseudolabel $\hat{Y}(x)$. Next, we use this graph and the empirical distribution of $\hat{Y}$ from $\bar{\mathcal{T}}_0$ to compute the $Z$-score for $x$. The last step is to threshold the $Z$-score to decide whether to set $\tilde{\mathcal{S}}(x) = 1$. We compute the quantiles of $Z$-score for the examples $\bar{\mathcal{T}}_0$, and we set $\tilde{\mathcal{S}}(x^{(0)}) = 1$ if $x$'s $Z$-score would have been in the top $\beta$ fraction in $\bar{\mathcal{T}}_0$. To summarize, to compute $\tilde{Y}(x^{(0)})$ we essentially perform the cut statistic on the set $\bar{\mathcal{T}}_0 \cup \{(x^{(0)}, x^{(1)}, \hat{Y}(x^{(0)}))\}$ (but where $x$ is left out of the percentile computations, and its $Z$-score is compared to the quantiles computed on $\bar{\mathcal{T}}_0$).

This sample-splitting trick allows us to extend the selection beyond the training sample to easily define a population-level $\tilde{Y}(X^{(0)})$:

$$\tilde{Y}(X^{(0)}) = \begin{cases} \hat{Y}(X^{(0)}) & \tilde{\mathcal{S}}(X^{(0)}) = 1 \\ \varnothing & \tilde{\mathcal{S}}(X^{(0)}) = 0 \end{cases}$$

We can now compute the relevant $\alpha$ and $\gamma$ parameters for $\tilde{Y}$, i.e., $\mathbb{P}[Y = 0 | \tilde{Y} = 1]$ and $\mathbb{P}[Y = 1 | \tilde{Y} = 0]$ are well-defined. Note that these parameters depend on $\bar{\mathcal{T}}_0$. We can then use $\bar{\mathcal{T}}_1$ in place of

$\bar{\mathcal{T}}$ as the full training data, and treat $\tilde{Y}$ like an arbitrary label model and applying Theorem 1. Note that the generalization bound now holds with respect to the sampling of $\bar{\mathcal{T}}_1$, while holding $\bar{\mathcal{T}}_0$ fixed. As a final check, observe that subset selection using $\mathcal{S}$ does not affect conditional independence: for $y \neq \varnothing$ and $B \subset \mathcal{X}^{(1)}$,

$$
\begin{aligned}
\mathbb{P}[X^{(1)} \in B, \tilde{Y}(X) = y | Y] &= \mathbb{P}[X^{(1)} \in B, \hat{Y}(X^{(0)}) = y, \mathcal{S}(X^{(0)}) = 1 | Y] \\
&= \mathbb{P}[X^{(1)} \in B | Y] \mathbb{P}[\hat{Y}(X^{(0)}) = y, \mathcal{S}(X^{(0)}) = 1 | Y] \\
&= \mathbb{P}[X^{(1)} \in B | Y] \mathbb{P}[\tilde{Y} | X].
\end{aligned}
$$

The first equality recalled that $\hat{Y}(X)$ and $\mathcal{S}(X)$ only depend on $\phi_0$, and the second used conditional independence of $X^{(0)}$ and $X^{(1)}$ given $Y$. The proof for $y = \varnothing$ is similar. Therefore, we can still apply Lemma 1 with $\tilde{Y}$. The samples $\mathcal{T}_1 = \{(x_i^{(1)}, \tilde{Y}(x_i^{(0)}))\}$ (which we obtained from $\bar{\mathcal{T}}_1$ by discarding $x_i^{(0)}$) are clearly i.i.d. because we assumed that $\{(x_i^{(1)}, x_i^{(0)})\}_{i=1}^n$ were i.i.d. samples. Hence, we can still apply the Theorem 2 result in the proof of Theorem 1. This sample-splitting construction of $\tilde{Y}$ allowed us to reduce to the case of a fixed, population level label model and directly use Theorem 1 to give bounds on the end model error when training with the refined pseudolabels.

### A.3 Balanced error generalization bound: Notation and result

The notation in this section is self-contained and slightly differs from that of previous sections. Let $\mathcal{X}$ be an input space and $\mathcal{Y} = \{0, 1, \varnothing\}$ be the (binary) label space + an abstention symbol. Let $\mathcal{H}$ be a class of functions mapping $\mathcal{X} \to \{0, 1\}$. We assume $(X, Y) \in \mathcal{X} \times \mathcal{Y}$ is a pair of random variables distributed according to an unknown distribution $\mathbb{P}$. We observe a sequence of $n$ i.i.d. pairs $(X_i, Y_i)$ sampled according to $\mathbb{P}$, and the goal is to learn a classifier $h \in \mathcal{H}$ with low *balanced error*:

$$
R(h) := \frac{1}{2} \left( \mathbb{P}[h(X) = 1 | Y = 0] + \mathbb{P}[h(X) = 0 | Y = 1] \right),
$$

To measure classifier performance from our finite sample, we use the *empirical balanced error*:

$$
R_n(h) := \frac{1}{2} \left( \frac{\sum_{i=1}^n \mathbb{1}_{h(X_i)=1} \mathbb{1}_{Y_i=0}}{\sum_{i=1}^n \mathbb{1}_{Y_i=0}} + \frac{\sum_{i=1}^n \mathbb{1}_{h(X_i)=0} \mathbb{1}_{Y_i=1}}{\sum_{i=1}^n \mathbb{1}_{Y_i=1}} \right)
$$

Note that the points where $Y = \varnothing$ do not appear in either expression. The goal is to derive a bound on the *generalization gap* $R(h) - R_n(h)$ for a classifier $\hat{h}$ that is learned from (and hence, depends on) the the finite sample $\{(X_i, Y_i) : i\}$. The challenge lies in the presence of random variables in the denominator of $R_n(h)$, so unlike the empirical zero-one loss, it cannot be written simply as $\frac{1}{n} \sum_{i=1}^n g(h, x, y)$ for some indicator function $g$.

The following theorem gives a uniform (over $\mathcal{H}$) convergence result for the balanced error. The key technique used in the proof is essentially due to Woodworth et al. [39].

**Theorem 2.** *For any $\delta \in (0, 1)$ and any distribution $P$, with probability at least $1 - \delta$ over the sampling of $\{(X_i, Y_i)\}_{i=1}^n$,*

$$
\sup_{h \in \mathcal{H}} R(h) - R_n(h) \leqslant \tilde{\mathcal{O}} \left( \sqrt{\frac{\mathrm{VC}(\mathcal{H}) + \log \frac{1}{\delta}}{n \mathbb{P}[Y \neq \varnothing] \min_{y \in \{0,1\}} \mathbb{P}[Y = y | Y \neq \varnothing]}} \right),
$$

*where $\tilde{\mathcal{O}}$ hides log factors in $n$ and $\mathrm{VC}(\mathcal{H})$.*

*Proof.* Let $S = \{(X_i, Y_i)\}_{i=1}^n$ refer to the empirical sample. For convenience, for each $(\bar{y}, y) \in \{0, 1\} \times \{0, 1\}$, define:

$$
\gamma_{\bar{y}y}(h) = \mathbb{P}[h(X) = \bar{y} | Y = y],
$$

and its empirical analogue:

$$
\gamma_{\bar{y}y}^S(h) = \frac{\sum_{i=1}^n \mathbb{1}_{h(X_i)=\bar{y}} \mathbb{1}_{Y_i=y}}{\sum_{i=1}^n \mathbb{1}_{Y_i=y}}.
$$

Then $R(h) = \frac{1}{2}(\gamma_{01} + \gamma_{10})$ and $R_n(h) = \frac{1}{2}(\gamma_{01}^S + \gamma_{10}^S)$. For sample $S$, let

$$
\mathcal{I}_y = \{i \in [n] : Y_i = y\}
$$

be the set of indices $i$ where $Y_i = y$, and set $n_y^S = |\mathcal{I}_y|$. Conditioned on $\mathcal{I}_y$, the $\gamma^S$ variables are distributed as:

$$\gamma_{\bar{y}y}^S(h)|\mathcal{I}_y \sim \frac{1}{n_y^S}\text{Binomial}(\gamma_{\bar{y}y}, n_y^S),$$

since the randomness over $X$ in the sample gives $n_y^S$ independent trials to make $h(X_i)$ equal to $\bar{y}$. We hide the argument $h$ below for convenience. Observe that $\mathbb{E}\left[\gamma_{\bar{y}y}^S \mid \mathcal{I}_y\right] = \gamma_{\bar{y}y}$. Then for every $\eta > 0$,

$$\mathbb{P}[|\gamma_{\bar{y}y}^S - \gamma_{\bar{y}y}| > t] = \sum_{\mathcal{I}_y} \mathbb{P}\left[|\gamma_{\bar{y}y}^S - \gamma_{\bar{y}y}| > t \mid \mathcal{I}_y\right]\mathbb{P}[\mathcal{I}_y]$$

$$\leqslant \mathbb{P}\left[n_y^S < (1-\eta)n\mathbb{P}[Y = y]\right] + \sum_{\mathcal{I}_y : n_y^S \geqslant (1-\eta)n\mathbb{P}[Y=y]} \mathbb{P}\left[|\gamma_{\bar{y}y}^S - \gamma_{\bar{y}y}| > t \mid \mathcal{I}_y\right]\mathbb{P}[\mathcal{I}_y]$$

$$\leqslant \exp\left(-\frac{\eta^2 n\mathbb{P}[Y = y]}{2}\right) + \sum_{\mathcal{I}_y : n_y^S \geqslant (1-\eta)n\mathbb{P}[Y=y]} 2\exp(-2n_y^S t^2)\mathbb{P}[\mathcal{I}_y]$$

$$\leqslant \exp\left(-\frac{\eta^2 n\mathbb{P}[Y = y]}{2}\right) + 2\exp(-2t^2(1-\eta)n\mathbb{P}[Y = y])$$

The first inequality comes from simplifying the sum over all $2^n$ possible values of $\mathcal{I}_y$. The second comes from applying a Chernoff bound to $\text{Binomial}(\mathbb{P}[Y = y], n)$ and Hoeffding's inequality to $\gamma_{\bar{y}y}$. We can set $\eta$ to balance these terms:

$$\frac{\eta^2}{2} = 2t^2(1-\eta),$$

which yields:

$$\eta = 2\left(\sqrt{t^4 + t^2} - t^2\right),$$

since the other root is negative. Substituting $\eta$ gives:

$$\mathbb{P}[|\gamma_{\bar{y}y}^S - \gamma_{\bar{y}y}| > t] \leqslant 3\exp\left(-2\left(\sqrt{t^4 + t^2} - t^2\right)^2 n\mathbb{P}[Y = y]\right)$$

For $t \in (0, 1)$, $\sqrt{t^4 + t^2} - t^2 \geqslant t/4$, so

$$\mathbb{P}[|\gamma_{\bar{y}y}^S - \gamma_{\bar{y}y}| > t] \leqslant 3\exp\left(-\frac{t^2}{8}n\mathbb{P}[Y = y]\right)$$

Hence, for $t \in (0, 1)$,

$$\mathbb{P}[|R(h) - R_n(h)| > t] \leqslant \mathbb{P}[|\gamma_{01} - \gamma_{01}^S| + |\gamma_{10} - \gamma_{10}^S| > 2t]$$
$$\leqslant \mathbb{P}[|\gamma_{01} - \gamma_{01}^S| > t] + \mathbb{P}[|\gamma_{10} - \gamma_{10}^S| > t]$$
$$\leqslant 6\exp\left(-\frac{t^2}{8}n\min_{y\in\{0,1\}}\mathbb{P}[Y = y]\right).$$
$$= 6\exp\left(-\frac{t^2}{8}n\mathbb{P}[Y \neq \varnothing]\min_{y\in\{0,1\}}\mathbb{P}[Y = y|Y \neq \varnothing]\right).$$

Now we show how to apply this deviation bound for $R$ in place of Hoeffding's inequality in the symmetrization argument from Bousquet et al. [4].

**Lemma 2** (Symmetrization). *Let $Z = (X, Y)$ and suppose we have a ghost sample of $n$ additional points $Z_i'$ drawn i.i.d. from $P$. Let $R_n'(h)$ denote the empirical balanced error of classifier $h$ on the ghost sample. Then for any $t > 0$ such that $nt^2 \geqslant \frac{32\log 12}{\min_{y\in\{0,1\}}\mathbb{P}[Y=y]}$:*

$$\mathbb{P}\left[\sup_{h\in\mathcal{H}} R(h) - R_n(h) > t\right] \leqslant 2\mathbb{P}\left[\sup_{h\in\mathcal{H}} R_n'(h) - R_n(h) > t/2\right].$$

*Proof of Lemma 2.* This follows Bousquet et al. [4] exactly, except we replace the application of one inequality with the deviation bound derived above. Let $h_n$ be the function achieving the supremum on the left-hand-side. This depends on the sample $(Z_1, \ldots, Z_n)$.

$$\mathbb{1}_{R(h_n)-R_n(h_n)>t}\mathbb{1}_{R(h_n)-R'_n(h_n)<t/2} = \mathbb{1}_{R(h_n)-R_n(h_n)>t \wedge R'_n(h_n)-R(h_n)\geqslant -t/2}$$
$$\leqslant \mathbb{1}_{R'_n(h_n)-R_n(h_n)>t/2}$$

Taking the expectation over the second sample $(Z'_1, \ldots, Z'_n)$,

$$\mathbb{1}_{R(h_n)-R_n(h_n)>t}\mathbb{P}'[R(h_n)-R'_n(h_n)<t/2] \leqslant \mathbb{P}'[R'_n(h_n)-R_n(h_n)>t/2]$$

From the result above,

$$P'[R(h_n)-R'_n(h_n)\geqslant t/2] \leqslant 6\exp\left(-\frac{t^2}{32}n\min_{y\in\{0,1\}}\mathbb{P}[Y=y]\right)$$
$$\leqslant \frac{1}{2}$$

by the condition on $nt^2$. Hence

$$\mathbb{1}_{R(h_n)-R_n(h_n)>t} \leqslant 2\mathbb{P}'[R'_n(h_n)-R_n(h_n)>t/2],$$

and taking the expectation over the original sample $(Z_1, \ldots, Z_n)$ finishes the proof. $\qquad\square$

Define $\mathcal{H}_{Z_1,\ldots,Z_n} = \{(h(x_1),\ldots h(x_n)) : h \in \mathcal{H}\}$. Recall that the *growth function* of class $\mathcal{H}$ is defined as $\mathcal{S}_{\mathcal{H}}(n) = \sup_{(Z_1,\ldots,Z_n)}|\mathcal{H}_{Z_1,\ldots,Z_n}|$. Now to finish the proof of Theorem 2, observe that the sup in the right-hand-side of the Lemma 2 result only depends on the *finite* set of vectors $\mathcal{H}_{Z_1,\ldots,Z_n,Z'_1,\ldots,Z'_n}$. That is,

$$\mathbb{P}\left[\sup_{h\in\mathcal{H}}R(h)-R_n(h)>t\right] \leqslant 2\mathbb{P}\left[\sup_{h\in\mathcal{H}_{Z_1,\ldots,Z_n,Z'_1,\ldots,Z'_n}}R'_n(h)-R_n(h)>t/2\right]$$
$$\leqslant 2\mathcal{S}_{\mathcal{H}}(2n)\max_{h\in\mathcal{H}_{Z_1,\ldots,Z_n,Z'_1,\ldots,Z'_n}}\mathbb{P}[R'_n(h)-R_n(h)>t/2]$$
$$\leqslant 4\mathcal{S}_{\mathcal{H}}(2n)\mathbb{P}[R(h)-R_n(h)>t/4]$$
$$\leqslant 24\mathcal{S}_{\mathcal{H}}(2n)\exp\left(-\frac{t^2}{128}n\min_{y\in\{0,1\}}\mathbb{P}[Y=y]\right),$$

where in the first line we applied the definition of the growth function and used the union bound, and in the last line we applied the concentration result for fixed $h$. Recall that the Sauer-Shelah lemma [38, 32, 36] implies that for any class $\mathcal{H}$ with $\mathrm{VC}(\mathcal{H}) = d$, $\mathcal{S}_{\mathcal{H}}(n) \leqslant \left(\frac{en}{d}\right)^d$. Then setting:

$$t \geqslant 8\sqrt{2\frac{\mathrm{VC}(\mathcal{H})\log\frac{2en}{\mathrm{VC}(\mathcal{H})}+\log\frac{24}{\delta}}{n\min_{y\in\{0,1\}}\mathbb{P}[Y=y]}}$$

completes the proof. Note that this choice of $t$ ensures that $nt^2 \geqslant \frac{32\log 12}{\min_{y\in\{0,1\}}\mathbb{P}[Y=y]}$ for any $\delta \in (0,1)$. $\qquad\square$

Table 4: Details for the WRENCH datasets used in this work.

| Task | Domain | Dataset | Num. Labels | # $\Lambda$'s | Train | Val | Test |
|------|--------|---------|-------------|--------------|-------|-----|------|
| Sentiment | Movie | IMDb | 2 | 5 | 20,000 | 2,500 | 2,500 |
| | Review | Yelp | 2 | 8 | 30,400 | 3,800 | 3,800 |
| Spam Classification | Comments | Youtube | 2 | 10 | 1,586 | 200 | 250 |
| Question Classification | Web Query | TREC | 6 | 68 | 4,965 | 500 | 500 |
| Relation Classification | Web Text | SemEval | 9 | 164 | 1,749 | 200 | 692 |
| | Chemical | ChemProt | 10 | 26 | 12,861 | 1,607 | 1,607 |
| | Biomedical | CDR | 2 | 33 | 8,430 | 920 | 4,673 |
| Image Classification | Video Frames | Basketball | 2 | 4 | 17,970 | 1,064 | 1,222 |
| Topic Classification | News | AGNews | 4 | 9 | 96,000 | 12,000 | 12,000 |

Table 5: Hyperparameter search spaces for label models and end models.

| Model | Parameters | Searched Values |
|-------|------------|-----------------|
| MeTaL | learning rate | 1e-5, 1e-4, 1e-3, 1e-2, 1e-1 |
| | weight decaay | 1e-5, 1e-4, 1e-3, 1e-2, 1e-1 |
| | training epochs | 5, 10, 50, 100, 200 |
| Data Programming | learning rate | 1e-5, 5e-5, 1e-4 |
| | weight decaay | 1e-5, 1e-4, 1e-3, 1e-2, 1e-1 |
| | training epochs | 5, 10, 50, 100, 200 |
| Logistic Regression | learning rate | 1e-5, 1e-4, 1e-3, 1e-2, 1e-1 |
| | weight decaay | 1e-5, 1e-4, 1e-3, 1e-2, 1e-1 |
| | batch size | 32, 128, 512 |
| | training steps | 10000 |
| MLP | learning rate | 1e-5, 1e-4, 1e-3, 1e-2, 1e-1 |
| | weight decaay | 1e-5, 1e-4, 1e-3, 1e-2, 1e-1 |
| | batch size | 32, 128, 512 |
| | training steps | 10000 |
| | hidden layers | 1 |
| | hidden size | 100 |
| BERT, RoBERTa | learning rate | 2e-5,3e-5,5e-5 |
| | weight decay | 1e-4 |
| | batch size | 16, 32 |
| | training steps | 10000 |

# B  All Empirical Results

## B.1  Dataset and hyperparameter details

Table 4 is a reproduction of Zhang et al. [42]'s Table 5 for the datasets used in this paper. Zhang et al. [42]'s Table 5 contains more statistics on the labeling functions, including average coverage and accuracy.

Table 5 shows the hyperparameter search spaces for the label models and end models. We used the same search spaces and tuning procedure as Zhang et al. [42] (see their Table 10), choosing the values that obtain the best mean performance on the gold-labeled validation set across five trial runs. As discussed in Section 4, we do *not* re-tune these hyperparameters for $\beta < 1.0$; we used fixed values to show that simply tuning $\beta$ on its own can improve performance.

## B.2  End model performance and $\beta$

Figure 6 shows how the end model test performance changes with $\beta$ for TREC and SemEval datasets and Majority Vote and Dawid-Skene label models. At low coverage fractions, the end model performs

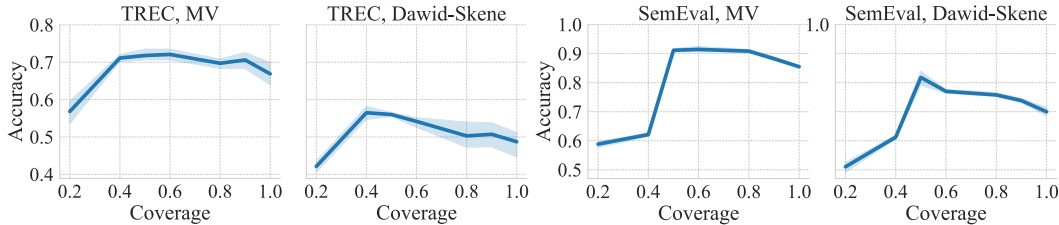

Figure 6: End model performance versus coverage fraction $\beta$ for TREC and SemEval, Majority Vote and Dawid-Skene.

worse than in the $\beta = 1.0$ case because there is less training data and because the training subsets can be imbalanced (recall that we do not use stratified subset selection, and that the generalization bound in Theorem 1 depends on the *minimum* coverage for each class). At the intermediate coverage fractions, the end model performs *better* than the $\beta = 1.0$ case. An interesting direction for future work is to determine methods for automatically selecting the best value of $\beta$.

### B.3 Subset selection versus relabeling

**Why not *correct* pseudolabels with nearest neighbor?**   Consider an example $x_i$ whose weak label $\hat{Y}(x_i)$ disagrees with the weak label $\hat{Y}(x_j)$ of most neighbors $j \in N(i)$. This example would get thrown out by the cut statistic selection. Instead of throwing such data points out, we could try to *re-label* them with the majority weak label from the neighbors. However, throwing data out is a more conservative (and hence possibly more robust) approach: Figure 7a shows a simple example where relabeling performs much worse than sub-selection. For the representations studied in this work, relabeling largely fails to improve training set quality and end model performance.

If the weak labels are mostly inaccurate close to the true unknown decision boundary (e.g., on hard examples), relabeling can actually make the training set *worse*. This is also borne out on real empirical examples. Figure 7b shows the weak label accuracy on a relabeled Yelp training set where the $\beta$ fraction of examples with the *largest* cut statistic score score $Z_i$—examples with many more cut edges than expected—are relabeled according to the majority vote of their neighbors. Relabeling largely fails to improve over the quality of the $\beta = 1.0$ full training set. However, we note that instead of relabeling, [6] obtained good results using nearest-neighbor to *expand* the pseudolabeled training set by labeling some of the unlabeled examples $\{x_i : \Lambda(x_i) = \varnothing\}$. If most uncovered examples $\{x : \hat{Y}(x) = \varnothing\}$ are closer to *correctly* pseudolabeled examples than incorrectly labeled ones, this nearest-neighbor *expansion* can improve performance.

### B.4 Additional selection accuracy plots

Figure 8 contain analogous plots to Figure 2 for every dataset in Table 1. These figures compare the quality of the training subsets selected by the cut statistic and entropy scoring.

### B.5 Ablation for number of cut statistic neighbors

This table shows how the results change when varying the number of nearest-neighbors $K$ used in the cut statistic, using majority vote and training a RoBERTa end model on TREC. The performance gain over $\beta = 1.0$ (which obtains 66.28% mean accuracy) is not sensitive to the choice of $K$. As in all of our results, we re-used hyperparameters from the $\beta = 1.0$ case and chose the best value of $\beta = 1.0$ according to gold-labeled validation performance. The best

| K value | Test accuracy |
| --- | --- |
| 5 | 76.92 (2.57) |
| 10 | 73.72 (2.59) |
| 20 | 72.92 (1.31) |
| 40 | 73.12 (2.39) |
| 80 | 72.16 (1.89) |

value for $\beta$ was stable across all choices of $K$: $\beta = 0.4$ had the optimal validation performance in all of these experiments. As indicated in the table, better results may be obtained by tuning over $K$, but our results showed the same value of $K$ obtains good performance across a wide variety of datasets and end models.

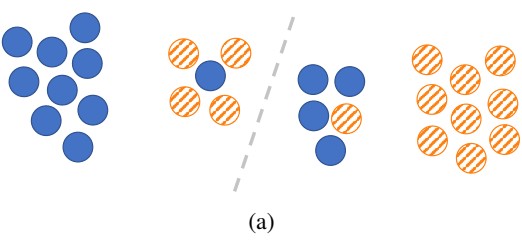
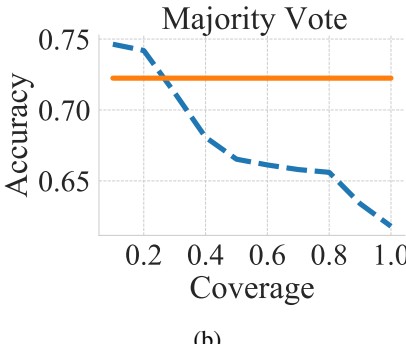

(a)

(b)

Figure 7: Left: example where subset selection performs better than re-labeling using $\phi$. In this example, the true labels $Y$ are recoverable by an unknown linear classifier (the dotted line). The weak labels (solid versus striped) are mostly incorrect close to this unknown decision boundary and always correct farther from the decision boundary. Relabeling the noisy points using nearest-neighbor (e.g., 4-nearest neighbor) actually makes the weak label accuracy *worse*, whereas selecting based on the cut statistic yields a subset of examples with 100% accuracy. Right: relabeling performance on Yelp. Points $(\beta, Y)$ show the accuracy of the pseudo-labeled training set obtained by relabeling the noisiest $\beta$ fraction of points (ranked by the cut statistic $Z_i$) with the majority vote of their neighbors in $\phi$ (dotted blue), compared to the accuracy when $\beta = 1$ (solid orange). Relabeling largely fails to improve accuracy over the $\beta = 1.0$ case.

Table 6: RoBERTa results using $\beta = 1.0$ (no sub-selection), reported from Zhang et al. [42]

| | imdb | yelp | youtube | trec | semeval | chemprot | agnews |
|---|---|---|---|---|---|---|---|
| MV [42] | 85.76 (0.70) | 89.91 (1.76) | 96.56 (0.86) | 66.28 (1.21) | 84.00 (0.84) | 56.85 (1.91) | 86.88 (0.98) |
| MV (ours) | 86.99 (0.55) | 88.51 (3.25) | 95.84 (1.18) | 67.60 (2.38) | 85.83 (1.22) | 57.06 (1.12) | 87.46 (0.53) |
| DP [42] | 86.26 (1.02) | 89.59 (2.87) | 95.60 (0.80) | 72.12 (4.58) | 70.57 (0.83) | 39.91 (9.33) | 86.81 (0.42) |
| DP (ours) | 86.31 (1.53) | 88.73 (5.07) | 94.08 (1.48) | 71.40 (3.30) | 71.07 (1.66) | 52.52 (0.69) | 86.75 (0.24) |
| DS [42] | 84.74 (1.41) | 92.30 (1.75) | 93.52 (1.39) | 48.32 (1.50) | 69.67 (1.18) | 45.69 (0.86) | 87.16 (0.58) |
| DS (ours) | 85.50 (1.68) | 92.42 (1.41) | 92.48 (1.44) | 51.24 (3.50) | 70.83 (0.75) | 45.61 (2.60) | 87.29 (0.40) |
| FS [42] | 86.95 (0.58) | 92.08 (2.63) | 93.84 (1.57) | 30.44 (3.48) | 31.83 (0.00) | 39.95 (6.50) | 86.69 (0.29) |
| FS (ours) | 85.25 (1.96) | 92.14 (2.76) | 93.52 (2.11) | 35.40 (1.32) | 31.83 (0.00) | 47.23 (1.04) | 86.56 (0.55) |
| MeTaL [42] | 84.98 (1.07) | 89.08 (3.71) | 94.56 (0.65) | 60.04 (1.18) | 70.73 (0.68) | 54.59 (0.77) | 87.18 (0.45) |
| MeTaL (ours) | 86.16 (1.13) | 88.41 (3.25) | 92.40 (1.19) | 55.44 (1.08) | 59.53 (1.87) | 56.74 (0.58) | 86.74 (0.60) |

## B.6 Original WRENCH results

Our $\beta = 1.0$ results closely matched the $\beta = 1.0$ results from Zhang et al. [42], but not in every case, despite using the same hyperparameter search space and tuning scheme for both the label model and the end model. Table 6 shows our results for RoBERTa and $\beta = 1.0$ in line with the same results reported in Zhang et al. [42]. Table 7 compares the performance of our method (i.e., selection with the cut statistic) against the performance of COSINE [41], which performs multiple rounds of self-training on the unlabeled data.

## B.7 Combining the Cut Statistic with Weakly-Supervised Self-Training Methods

**COSINE.** COSINE [41] combines an initial set of pseudolabeled data with a self-training procedure to make better use of the *unlabeled* data that is not covered by weak rules. In each round of self-training, a subset of the data is chosen to use as the training set for the next round by using the confidence score of the trained end model. Instead of using the confidence score, we can instead use the cut statistic to select the training data for each round. This is analogous to switching from the standard self-training algorithm to SETRED [19], which uses the cut statistic to select data for each self-training round. Intuitively, replacing the poorly-calibrated confidence score with the cut statistic

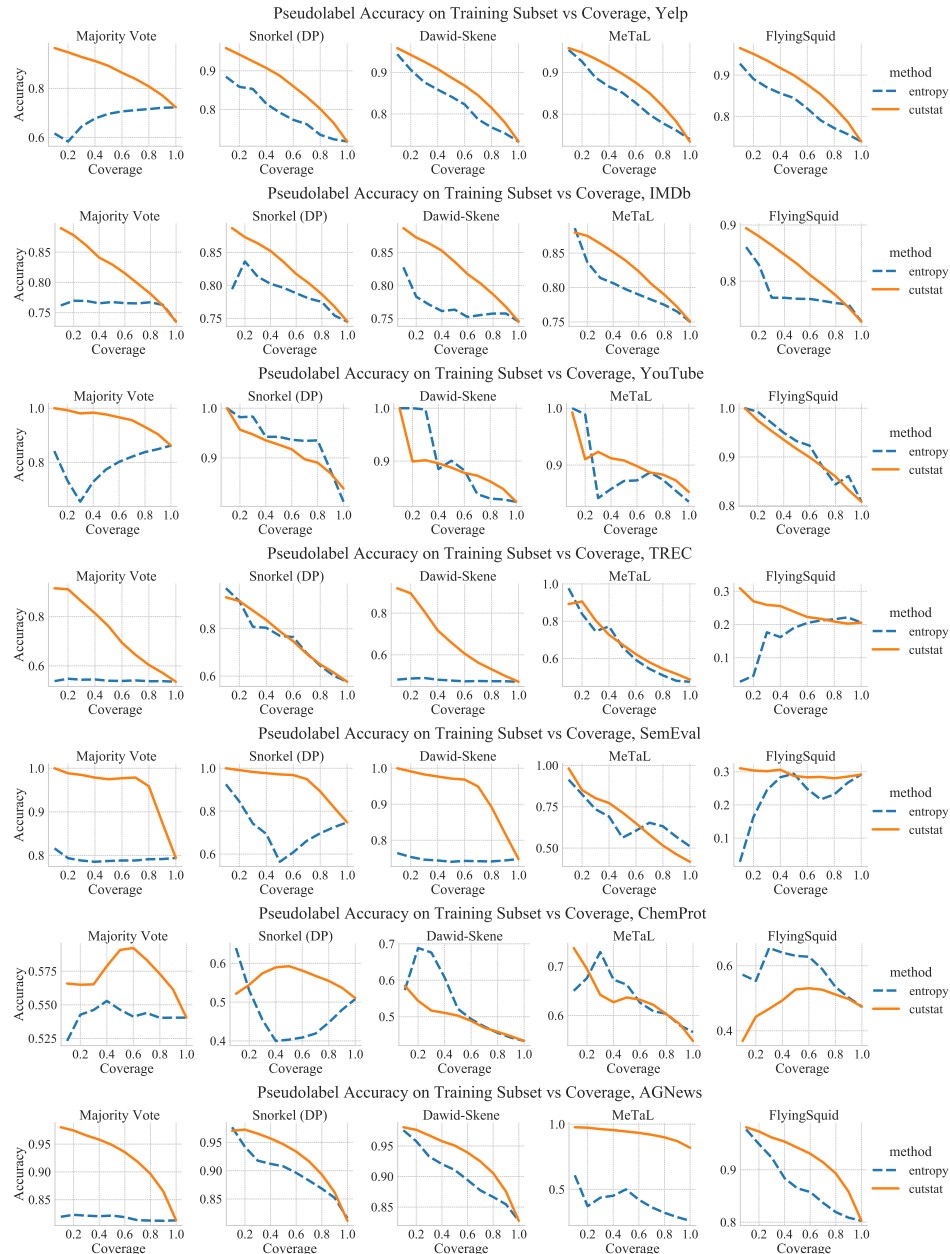

Figure 8: Accuracy of the pseudolabeled training set versus the selection fraction $\beta$ for five different label models and seven datasets. A pretrained BERT model is used as $\phi$ for the cut statistic. The accuracy of the weak training labels is better for $\beta < 1$, indicating that sub-selection can select higher-quality training sets. The two curves should always agree at $\beta = 1.0$, but don't always do so for MeTaL due to noise in the MeTaL training procedure.

in each round should lead to higher quality training data and increased performance. Our previous experiments show this is true for the *first* round.

**ASTRA.** ASTRA [15] is a semi-weakly supervised learning method that uses weakly-labeled data plus a small set of labeled data and a large set of unlabeled data. There are two networks: the *teacher model*, also called the Rule Attention Network (RAN), and the *student model*, which is analogous to our *end model* (BERT, RoBERTa, etc.). Training proceeds in rounds. In the first step, the student model is fine-tuned on the small labeled dataset and used to pseudolabel the unlabeled

Table 7: Cut statistic versus the COSINE method [41]. We report the tuned COSINE performance from [42]. COSINE performs multiple rounds of self-training on the unlabeled data, whereas the cut statistic method performs one round of training on a carefully chosen subset of the weakly-labeled data. Surprisingly, the cut statistic is sometimes competitive with COSINE despite not using any of the unlabeled data and only requiring one round of training. We show in Section B.7 how to combine two appraoches.

| End Model | Method | imdb | yelp | youtube | trec | semeval | chemprot | agnews |
|-----------|--------|------|------|---------|------|---------|----------|--------|
| BERT | MV + COSINE | 82.98 (0.05) | 89.22 (0.05) | 98.00 (0.00) | 76.56 (0.08) | 86.80 (0.46) | 58.47 (0.08) | 87.03 (0.00) |
| | MV + cutstat | 81.86 (1.36) | 89.49 (0.78) | 95.60 (0.72) | 71.84 (3.00) | 92.47 (0.49) | 57.47 (1.00) | 86.26 (0.43) |
| | DP + COSINE | 84.58 (0.08) | 88.44 (0.03) | 96.32 (0.16) | 78.72 (0.43) | 75.77 (1.33) | 57.51 (0.02) | 86.98 (0.39) |
| | DP + cutstat | 79.07 (2.52) | 88.13 (1.46) | 93.92 (0.93) | 76.76 (1.92) | 91.07 (0.90) | 55.10 (1.49) | 85.89 (0.45) |
| | DS + COSINE | 91.54 (0.54) | 90.84 (0.30) | 94.16 (0.20) | 53.36 (0.29) | 72.50 (0.00) | 49.65 (0.68) | 87.19 (0.00) |
| | DS + cutstat | 80.22 (1.69) | 89.04 (1.10) | 90.72 (1.27) | 57.28 (2.91) | 89.07 (1.62) | 49.07 (1.48) | 86.93 (0.22) |
| | FS + COSINE | 84.40 (0.00) | 89.05 (0.07) | 94.80 (0.00) | 27.60 (0.00) | 31.83 (0.00) | 48.10 (0.60) | 87.16 (0.16) |
| | FS + cutstat | 80.85 (1.50) | 88.75 (1.13) | 91.04 (1.23) | 33.84 (3.17) | 31.83 (0.00) | 48.65 (0.99) | 85.90 (0.39) |
| | MeTaL + COSINE | 83.47 (0.12) | 89.76 (0.00) | 94.88 (0.53) | 61.80 (0.00) | 79.20 (2.33) | 55.46 (0.12) | 87.26 (0.02) |
| | MeTaL + cutstat | 81.49 (1.51) | 88.41 (1.19) | 92.64 (0.41) | 63.80 (2.28) | 65.23 (0.91) | 58.33 (0.81) | 86.16 (0.48) |
| RoBERTa | MV + COSINE | 88.22 (0.22) | 94.23 (0.20) | 97.60 (0.00) | 77.96 (0.34) | 86.20 (0.07) | 59.43 (0.00) | 88.15 (0.30) |
| | MV + *cutstat* | 86.69 (0.75) | 95.19 (0.23) | 96.00 (1.10) | 72.92 (1.31) | 92.07 (0.80) | 59.05 (0.56) | 88.01 (0.47) |
| | DP + COSINE | 87.91 (0.15) | 94.09 (0.06) | 96.80 (0.00) | 82.36 (0.08) | 75.17 (0.95) | 52.86 (0.06) | 87.53 (0.03) |
| | DP + *cutstat* | 86.46 (1.82) | 93.95 (0.93) | 93.04 (1.30) | 76.84 (4.09) | 86.07 (1.82) | 56.43 (1.37) | 87.76 (0.17) |
| | DS + COSINE | 88.01 (0.56) | 94.19 (0.18) | 96.24 (0.41) | 59.40 (0.42) | 71.70 (0.07) | 46.75 (0.27) | 88.20 (0.11) |
| | DS + *cutstat* | 86.14 (0.60) | 93.81 (0.69) | 93.84 (0.70) | 58.48 (2.75) | 81.67 (1.33) | 52.93 (1.67) | 88.35 (0.22) |
| | FS + COSINE | 88.48 (0.00) | 95.33 (0.06) | 96.80 (0.00) | 33.80 (0.00) | 31.83 (0.00) | 39.89 (0.00) | 87.23 (0.00) |
| | FS + *cutstat* | 87.71 (0.76) | 94.50 (0.74) | 95.84 (0.54) | 38.16 (0.43) | 31.83 (0.00) | 50.55 (1.05) | 87.49 (0.13) |
| | MeTaL + COSINE | 86.46 (0.11) | 93.11 (0.01) | 97.04 (0.20) | 71.64 (0.59) | 70.90 (0.08) | 53.32 (0.19) | 87.85 (0.02) |
| | MeTaL + *cutstat* | 87.46 (0.65) | 94.03 (0.53) | 93.84 (1.38) | 69.72 (2.39) | 66.70 (0.90) | 57.40 (0.98) | 88.40 (0.38) |

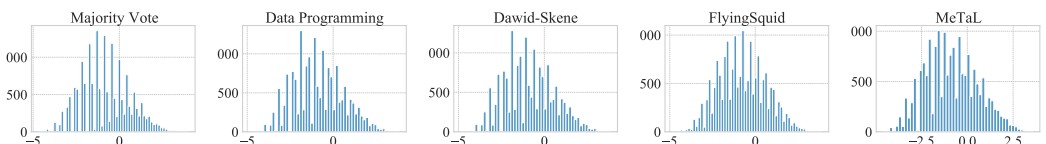

Figure 9: Histograms for the cut statistic score $Z_i$ on IMDb using BERT as $\phi$.

dataset. Next, the teacher is trained on the weakly-labeled training data plus pseudolabeled data from the gold-fine-tuned student. The teacher then pseudolabels the unlabeled data using a learned instance-specific weighting procedure, so that high-quality examples are upweighted. Finally, the student model is trained on this data.

We can insert the cut statistic in multiple parts of this procedure. First, in each round, ASTRA trains the teacher model on *all* of the pseudolabeled data from the gold-fine-tuned student. Instead, we can use the cut statistic to select a high-quality subset of this data for the teacher model to train on. Second, the student model is trained on a subset of the data selected by the teacher model; we can further filter this subset with the cut statistic. Applying the cut statistic in this step is somewhat less necessary, since ASTRA already has a (soft) instance-specific selection procedure built-in.

Table 8 shows plain ASTRA versus ASTRA + cutstat on SemEval and TREC using a RoBERTa-base end model. Standard deviations are reported across five random seeds for choosing the labeled subset. Following the best constant $\beta$ from Section 4, we set $\beta = 0.6$ for the first round of training, then increase by 0.1 in each round to use more of the unlabeled data each time. So $\beta$ for the $t$-th round of ASTRA is $\min(1, 0.6 + 0.1t)$, $t \in \{0, \ldots, 24\}$. Following Karamanolakis et al. [15], we train ASTRA for up to 25 iterations using a patience of 3 iterations. In each step, the model checkpoint with best validation performance is kept. We did not perform hyperparameter tuning on the end model parameters and used a fixed learning rate of 2e-5 and batch size 128. The cut statistic improves the ASTRA performance for nearly every labeled data size despite us not tuning $\beta$ on the validation set. Tuning $\beta$ on the validation set, as in Table 1, would likely result in even better performance gains.

Table 8: Combining the cut statistic with ASTRA [15] boosts performance by selecting a higher-quality set of training data for the teacher model in each round. These results use fixed end-model hyperparameters and a fixed choice for the cut statistic fraction $\beta$ in each round.

| Method | \|Labeled set\| | trec | semeval |
|---|---|---|---|
| ASTRA | 10 | 65.60 (5.19) | 82.70 (3.04) |
| + *cutstat* | 10 | 67.40 (5.78) | 91.10 (0.92) |
| ASTRA | 20 | 74.40 (3.35) | 86.53 (1.17) |
| + *cutstat* | 20 | 75.04 (1.63) | 90.27 (2.09) |
| ASTRA | 40 | 85.72 (1.32) | 87.60 (1.22) |
| + *cutstat* | 40 | 84.52 (3.17) | 91.20 (1.11) |

## C  Cut statistic code

For simplicity in computing the graph $G$ for the cut statistic, we provide code where the neighborhoods sets $N(i)$ are not necessarily symmetric, so $i \in N(j) \implies j \in N(i)$. This does not change the empirical performance of the algorithm.

```python
import torch

def get_conf_inds(labels, features, coverage, device='cuda'):
    features = torch.FloatTensor(features).to(device)
    labels = torch.LongTensor(labels).to(device)

    # move to CPU for memory issues on large dset
    pairwise_dists = torch.cdist(features, features, p=2).to('cpu')

    N = labels.shape[0]
    dists_sorted = torch.argsort(pairwise_dists)
    neighbors = dists_sorted[:,:20]
    dists_nn = pairwise_dists[torch.arange(N)[:,None], neighbors]
    weights = 1/(1 + dists_nn)

    neighbors = neighbors.to(device)
    dists_nn = dists_nn.to(device)
    weights = weights.to(device)

    cut_vals = (labels[:,None] != labels[None,:]).long()
    cut_neighbors = cut_vals[torch.arange(N)[:,None], neighbors]
    Jp = (weights * cut_neighbors).sum(dim=1)

    weak_counts = torch.bincount(labels)
    weak_pct = weak_counts / weak_counts.sum()

    prior_probs = weak_pct[labels]
    mu_vals = (1-prior_probs) * weights.sum(dim=1)
    sigma_vals = prior_probs * (1-prior_probs) * torch.pow(weights, 2).sum(dim=1)
    sigma_vals = torch.sqrt(sigma_vals)
    normalized = (Jp - mu_vals) / sigma_vals

    normalized = normalized.cpu()
    inds_sorted = torch.argsort(normalized)

    N_select = int(coverage * N)
    conf_inds = inds_sorted[:N_select]
    conf_inds = list(set(conf_inds.tolist()))
    return conf_inds
```