# OpenReview forum: "Training Subset Selection for Weak Supervision"
_NeurIPS.cc/2022/Conference — NeurIPS 2022 Accept_

### Official Review · Reviewer_Y4et · 2022-07-09

**Rating:** 5
**Confidence:** 4
**Soundness:** 3 good
**Presentation:** 3 good
**Contribution:** 2 fair

**Summary:**

This paper proposes a simple dataset selection strategy for weakly-supervised
learning. The strategy is based on the cut statistics, which constructs a
similarity graph among the instances and select the nodes that have consistent
labels with their neighbors. The authors have shown such a simple data
selection strategy enhances previous labeling models for weakly-supervised
learning. They also provide a theoretical analysis on the generalization
bound, which formulates a tradeoff between precision and coverage for weak
labels.


**Questions:**


Please see the weaknesses above.

Besides the above two concerns, an additional question is that there are
recent work [1] showing relabeling is better than discarding data in the
context of supervised learning. It'll be great if the authors can comment on
these two seemingly contradictory findings.

[1] Resolving Training Biases via Influence-based Data Relabeling



**Limitations:**

Yes, the authors have adequately addressed the limitations and potential negative societal impact.


**Strengths And Weaknesses:**


Strengths:

S1. The proposed selection strategy based on cut statistics is simple but effective empirically.

S2. The authors have conducted experiments on datasets with different modalities, including text, image, and tabular data.

S3. The paper is well written and easy to follow.


Weaknesses:

W1. There is a gap between the theorem and the experiments. In the theorem,
the author assumes two different views that are conditionally independent. The
labeling functions are defined on one view, and the classifier is trained on
the other. Unless I'm missing anything, there are no such complementary views
on the datasets used in experiments. Also, even if we have multi-view data
sets in practice, the assumption that the two views are conditionally
independent, and that we use the two view separately for labeling function and
model training are impractical.

W2. The authors mainly evaluated the selection strategy for labeling model.
However, as the authors mentioned, there are works (e.g., cosine) that perform
data selection during the end model training. I do not see any major obstacles
that prevent the proposed data selection strategy from being applied during
the end model training. It'll be more interesting to see how this measure compared with the previous methods that uses data selection during the end model training.

---

> ### Author Response · Authors · 2022-08-02
> **Response to Reviewer Y4et**
>
> Thanks for your time and helpful feedback! We respond to your individual points below.
>
> > if we have multi-view data sets in practice, the assumption that the two views are conditionally independent, and that we use the two view separately for labeling function and model training are impractical.
>
> Thanks for pointing this out! Our theoretical setup is motivated by a common application of weak supervision: chest X-ray datasets. The label for X-ray images in the CheXPert and MIMIC-CXR datasets are derived from weak supervision over text-based notes. The notes are not available to the final classifier both due to privacy reasons and because the actual goal is to train a classifier for *images only* that can still operate before notes are even written. So the multimodal case of using two separate views, and only using one view for classification, does have notable applications. We tried to mention this in Section 5 but we would improve the description in the extra space allowed by the camera ready based on your feedback.
>
> Additionally, the notes typically just describe the conditions (i.e., the *labels*) present in the Xray, so conditional independence between the notes and the images given the true label might not be that far-fetched. But we agree that strict conditional independence given the label is implausible.
>
> As you point out, all the datasets we study have only one view. Giving theoretical guarantees for weak supervision in this setting is very difficult because the weak labels only emit on a biased subset of the data distribution. Why does a classifier trained on this covered subset extend to the uncovered set? We are actively pursuing further theoretical research on this topic.
>
> Our main focus in this paper was to give a very simple (< 30 lines of code) and efficient method for boosting the performance of two-stage weak supervision methods across the board. Our empirical findings in Tables 1 and 2 show a large and robust effect.
> The purpose of Theorem 1 is to indicates that the technique has some theoretical footing in a specific setting. It shows that there is a tradeoff between precision and coverage for weak rules that is intuitive, but has not been explored before in the weak supervision literature. Theorem 1 is also novel because it shows a case where very tight end-model generalization bounds exist. Prior work on end-model generalization (e.g., reference [10]) has only given bounds that are effectively of the form
> $$err(\text{end model})\le err(\text{optimal label model}) + \text{positive term}$$
> But in practice, the end model outperforms the optimal label model.
> In contrast to those prior bounds, the bound in Theorem 1 goes to 0 with the amount of weakly-labeled data.
>
> In summary, we view our main contribution as the strong and robust empirical effect of our very simple method. Theorem 1 is meant to bring theoretical attention to the tradeoff between precision and coverage and to give a tighter generalization bound than those in the existing literature. Based on your feedback, we will update the paper to better contextualize the novelty of Theorem 1 compared to other weak supervision bounds and to justify the assumptions in more detail.
>
> > interesting to see how this measure compared
>   with the previous methods that uses data selection during the end
>   model training.
>
> Thanks for this suggestion. As mentioned in our response to Reviewer TRjo, we will update the paper to include a comparison to the performance of COSINE. Our method sometimes beats COSINE despite only training the end model once and not using the unlabeled data at all (c.f. Table 12 in the WRENCH paper, Zhang et al. 2021, and our response to Reviewer TRjo).
> However, we do not view self-training approaches like COSINE and ASTRA as competing methods to the cut statistic: they can be easily combined, as mentioned in our response to TRjo. We plan to include a discussion of this combination.
>
> >  recent work [1] showing relabeling is better than discarding data in the context of supervised learning. It'll be great if the authors can comment on these two seemingly contradictory findings.
>
> Thanks for pointing us to this interesting paper! We believe the difference is due to the nature of the noise and the method used for relabeling. We considered relabeling using the cut statistic (essentially, nearest-neighbor relabeling of the points with noisiest neighborhoods). Figure 7 in the Supplementary material shows a simple example where relabeling with the cut statistic performs worse than just throwing out the noisy points. Note the structure of the noise in this example.
>
> We believe a similar example may cause influence-function-based relabeling to break. The empirical results in that paper seem to use a very special independent-noise model, whereas our experiments deal with more realistic noise in the weak labels.

---

> > ### Author Response · Authors · 2022-08-09
> > **Did we answer your concerns?**
> >
> > Thank you again for your comments and feedback on our manuscript. Since we are quickly approaching the end of the Author-Reviewer Discussion period, could you please let us know if you have any remaining concerns about the paper that we can address? If you think we have addressed your concerns, please consider raising your overall rating for the paper, if possible.

---

### Official Review · Reviewer_ocdh · 2022-07-11

**Rating:** 8
**Confidence:** 4
**Soundness:** 4 excellent
**Presentation:** 4 excellent
**Contribution:** 4 excellent

**Summary:**

This paper studies the subset selection problem of weakly supervised learning. By dividing a weakly supervised learning pipeline into labeling functions, label model and end model, the paper specifically studies on label model which motivates the proposed subset selection method based on cut-statistics. Different from entropy-based subset selection, cut statistic based selection relies on identifying highly noisy examples based on their proximity in a constructed feature graph. The proposed selection method serves as a plugin for any label model and empirical evaluations are presented to show that it improves many existing label model and end models.

**Questions:**

A minor question regarding connection to semi-weakly supervised learning: to get best performance on a particular task, a validation set is required to set the selection threshold \beta, then this essentially leverages information about true labels, I wonder how does the proposed method compare to weak label re-weighting / correction method. e.g., [1][2] (some of these require a small set of clean labels for learning, and for the studied setting in this paper weak labels + validation data, for a stretched setting we can split the validation data to use partially for training and the rest for validation and thus apply the above mentioned methods.)

[1] Meta-Weight-Net: Learning an Explicit Mapping For Sample Weighting. NeurIPS '19
[2] Meta Label Correction for Noisy Label Learning. AAAI '21

**Limitations:**

Potential limitations are well addressed.

**Strengths And Weaknesses:**

Strengths:

1. Using the cut statistics to identify highly noisy labels is an elegant and novel idea, which doesn't require any knowledge about true labels. Justifying using less data may actually lead to better performance in weakly supervised learning carries the most contribution to me.
2. The proposed selection approach is simple but yet very effective which improves many existing label model and end model.
3. The paper is extremely well presented ranging from motivation, formulation, method coverage, empirical and theoretical justifications.

Weakness:
I find no obvious weakness of the paper and it's a very enjoyable read.

---

> ### Author Response · Authors · 2022-08-02
> **Response to Reviewer ocdh**
>
> First, thanks for your positive comments about our paper!
>
> > for a stretched setting we can split the validation data to use partially for training and the rest for validation and thus apply the above mentioned methods.
>
> This is a great point and definitely an issue with current evaluation in the weak supervision literature. Assuming we have access to a decently large validation set, we might as well use a handful of examples as labeled training data.
> There is likely a "crossover point" where the benefit of using the examples for training is outweighed by the additional noise in the validation set, causing model/hyperparameter selection to be too noisy and degrade test performance.
> We have not seen any weak supervision papers explore this tradeoff.
>
> We believe the tradeoff is likely very dataset-specific. For example, for the YouTube dataset, there is effectively no benefit to seeing gold labeled examples during training: weak supervision with no labeled data essentially matches the fully-supervised performance from the start.
> For TREC, on the other hand, using 40 labeled examples from the validation set for training allows ASTRA (a semi-weakly-supervised method from  Karamanolakis et al., 2021) to comfortably outperform two-stage methods. ASTRA with RoBERTa-base obtains test accuracy 83.89 (stddev 3.41 across 10 random choices of the labeled set), compared to 76.84 (stddev 4.09 across end model initializations) for the best two-stage method (DP + cut statistic) and 71.40 (3.30) for the best two-stage method with no cut statistic. (However, as we mentioned in our response to another reviewer, the cut statistic can also be used to boost the performance of this approach.)
>
> This exact issue has come up recently in the few-shot learning literature for large language models (Perez et al. 2021). Recent works in that space such as Gao et al. (2021) sample a *small* validation set (around the same number of labeled examples used in the training set) and evaluate the performance of their methods across random draws of that set. Some other works in this space either use *weakly*-labeled validation data for selection (Lang et al. 2022) or don't perform hyperparameter tuning at all (Liu et al. 2022).
>
> Methods like ASTRA may be more sensitive to validation set sampling than two-stage methods like Snorkel, since in each self-training iteration they keep the best model checkpoint for use in the next iteration. A noisy validation set can cause the errors of a model to propagate, whereas the two-stage weak supervision approaches might be more robust since they only train the end model once. Hence, if the validation set is small, it might be better to go with a two-stage approach and no labeled data instead of siphoning off labeled examples for training. But we haven't empirically tested this hypothesis.
>
> We will add a brief discussion of this issue to the paper. We can also re-generate some of Table 1 using random draws of very small validation sets. It would be very interesting to do a large-scale comparison of different weak supervision methods with small, randomly-sampled validation sets rather than the sometimes large ones used in existing benchmarks (e.g., two-stage method + 40 val samples versus ASTRA with 20 train and 20 val). We think that is somewhat out-of-scope for this paper, since it would really be a comparison between the basic two-stage methods and semi-weakly-supervised methods like ASTRA, whereas the cut statistic can be used to boost the performance of both. But we think this is a very important regime to include in future weak supervision benchmarks. Thanks again for bringing this up, and for your helpful review!
>
> Gao, Tianyu, Adam Fisch, and Danqi Chen. "Making Pre-trained Language Models Better Few-shot Learners." Proceedings of the 59th Annual Meeting of the Association for Computational Linguistics and the 11th International Joint Conference on Natural Language Processing (Volume 1: Long Papers). 2021.
>
> Lang, Hunter, et al. "Co-training improves prompt-based learning for large language models." International Conference on Machine Learning. PMLR, 2022.
>
> Liu, Haokun, et al. "Few-shot parameter-efficient fine-tuning is better and cheaper than in-context learning." arXiv preprint arXiv:2205.05638 (2022).

---

> > ### Comment · Reviewer_ocdh · 2022-08-08
> > **thanks for the info**
> >
> > I am glad to see that you mentioned the trade-off between the benefit of using (partially) validation set for training and test performance. This is indeed a not well explored area for both unsupervised and weakly supervised learning. Looking forward to the additional results on using smaller validation sets.

---

### Official Review · Reviewer_APNJ · 2022-07-11

**Rating:** 6
**Confidence:** 5
**Soundness:** 3 good
**Presentation:** 3 good
**Contribution:** 3 good

**Summary:**

The authors propose a simple method for combining pretrained data representations with the cut statistic to select better quality subsets of the weakly-labeled training data. By selecting subsets of data rather than using the full covered data, a weak supervision label model is able to achieve better precision. The authors also show that their method also improves performance of end models.


**Questions:**

I like the direction of this work and the potential gain to weak supervision however I think a major flaw in the paper lies in selecting a good value for \beta. In the paper, the authors use a validation set to select the best \beta for the results in the table. This will not be practical when a validation set is not available and if the cost of labeling the data to obtain a validation set is high. Looking at the plots, there is no obvious alternative strategy for selecting \beta as it depends on the classification task. I think it will be very helpful if the authors can suggest an approach for selecting \beta when there is no access to validation data and maybe show experiments for it.

**Limitations:**

The authors addressed potential limitations of their paper. The work has no significant negative societal impact.

**Strengths And Weaknesses:**

Strengths:

1. The method is simple and easy to integrate with any weak supervision approach.

2. The authors show strong empirical results and provide theoretical guarantees for their approach.

Weakness:
 - Selecting good value of \beta may not be practical

---

> ### Author Response · Authors · 2022-08-02
> **Response to Reviewer APNJ**
>
> >  Selecting good value of \beta may not be practical
>
> Thanks for your feedback! Assuming access to a (possibly small) gold-labeled validation set is very common in the weak supervision literature.
> Many prior works (e.g., our references [3, 6, 10, 25, 26, 27, 38, 39]) use such a set to select end-model hyperparameters like learning rate, weight decay, batch size, etc., to select the best model checkpoint, and to select hyperparameters specific to the weak supervision method.
> We follow these works, and our method only has one more hyperparameter ($\beta$) than the usual two-stage weak supervision methods.
>
> As we discuss in the conclusion, we believe it's very important to have a gold-labeled test set to evaluate classifiers trained via weak supervision. Because the weak label coverage is biased (i.e., the set of data covered by weak rules != the full set of data), this is even more important than in the fully-supervised case. So in a hypothetical setting where we need to collect labeled test data anyway, it seems reasonable to use some of it for validation.
>
> That being said, choosing a constant $\beta=0.6$ still has reasonable average performance. For the 70 trials in Table 1, the median improvement of $\beta=0.6$ over $\beta=1.0$ is 1.7% (absolute) accuracy points. The mean improvement is 2.1% absolute. We will add a brief suggestion that this can be used as a relatively safe choice for beta if no validation data is available.
> We hope this clears up your concern about selecting $\beta$.

---

> > ### Author Response · Authors · 2022-08-09
> > **Did we answer your concerns?**
> >
> > Thank you again for your comments and feedback on our manuscript. Since we are quickly approaching the end of the Author-Reviewer Discussion period, could you please let us know if you have any remaining concerns about the paper that we can address? If you think we have addressed your concerns, please consider raising your overall rating for the paper, if possible.

---

### Official Review · Reviewer_TRjo · 2022-07-12

**Rating:** 7
**Confidence:** 4
**Soundness:** 3 good
**Presentation:** 3 good
**Contribution:** 3 good

**Summary:**

This paper proposes a method to improve the precision of existing weak supervision approaches by selecting a subset of the weakly-labeled data for training the end model. Specifically, the paper proposes to use the cut statistic that considers vector representations of the input examples, and keeps examples with the same pseudolabel as their nearest neighbors in this vector space. These examples are identified by constructing a graph where each instance is a node, computing cuts using weak labels, and identifying the number of cut edges for each node.

The proposed approach is evaluated across several benchmarks for weak supervision and compared with an entropy-based measure for subset selection.

**Questions:**

Are the authors planning to share their code for replicating the experiments reported in the paper (in addition to the code snippet in the appendix)?

**Limitations:**

Yes.

**Strengths And Weaknesses:**

Strengths:
* The paper is very clearly written and easy to follow.
* The paper proposes a simple method that effectively improves the precision of existing weak supervision methods across several datasets. It is a great benefit that the proposed method can be applied in general across label models, end models, and data modalities, and as a result, can serve as a useful tool for the weak supervision community.
* Additional experiments, such as evaluating the choice of the representation for the cut statistic are very interesting.
* The paper provides a theoretical analysis on a special case of weak supervision that motivates subset selection and can help better understand the precision-coverage tradeoff.

Weaknesses:
* The paper experimentally compares the cut statistic approach to just a very simple entropy-based approach and does not consider existing work that attempts to improve the precision of the existing sources, e.g., via instance-specific weights (Awasthi et al., 2020; Karamanolakis et al., 2021; Yu et al., 2021). A more thorough experimental comparison would better place the paper in the context of state-of-the-art approaches and would make the paper's contribution stronger.

Minor comments:
* Section 3.1 could be a better fit under the "Experiments" section.

References:
* Abhijeet Awasthi, Sabyasachi Ghosh, Rasna Goyal, and Sunita Sarawagi, "Learning from rules generalizing labeled exemplars", (ICLR 2020)
* Giannis Karamanolakis, Subhabrata Mukherjee, Guoqing Zheng, Ahmed Hassan Awadallah, "Self-training with weak supervision", (NAACL 2021)
* Yue Yu, Simiao Zuo, Haoming Jiang, Wendi Ren, Tuo Zhao, Chao Zhang, "Fine-Tuning Pre-trained Language Model with Weak Supervision: A Contrastive-Regularized Self-Training Approach", (NAACL 2021)

---

> ### Author Response · Authors · 2022-08-02
> **Response to Reviewer TRjo**
>
> Thanks for your detailed review and helpful feedback!
>
> > Are the authors planning to share their code for replicating the experiments reported in the paper (in addition to the code snippet in the appendix)?
>
> Yes, we plan to share all the code for reproducing our experiments.
> Our code is built on top of the open-source WRENCH library for benchmarking weak supervision methods. We will release the code used to generate the numbers in Table 1, and we also plan to submit a pull request to incorporate the cut statistic into WRENCH.
>
> >  does not consider existing work that attempts to improve the precision of the existing sources, e.g., via instance-specific weights (Awasthi et al., 2020; Karamanolakis et al., 2021; Yu et al., 2021). A more thorough experimental comparison would better place the paper in the context of state-of-the-art approaches and would make the paper's contribution stronger.
>
> Thanks for bringing up these methods! In this paper we wanted to focus on the simplest weak supervision approach: two-stage methods that only train the end model once. As far as we can tell, these are the most common in practice (e.g., the Snorkel repo has 5.2k stars on GitHub, MeTAL 409, FlyingSquid 288, versus 130 for ASTRA and 150 for COSINE).
> The advantage of two-stage approaches is that they only require training the end model once and don't require any labeled data.
> Methods like COSINE (Yu et al.) and ASTRA (Karamanolakis et al.) rely on multiple rounds of self-training and have more hyperparameters and more complex training. Additionally, ASTRA assumes access to some labeled examples.
>
> We view our main contribution as a very simple method for robustly improving two-stage approaches across the board. We believe the robustness of the results in Table 1 suggests this approach can be effective in practical, real-world applications.
>
> To better contextualize our results, we will add a comparison between the numbers in Table 1 to the COSINE performance reported in the WRENCH Benchmark (Zhang et al., Table 12). Note that by comparing to Table 12 in Zhang et al., in some cases, we beat COSINE despite only training the end model once and not using any of the unlabeled data.
> For example, COSINE + BERT with Majority Vote on SemEval obtains 86.80 (0.46). BERT + cutstat with Majority Vote on SemEval obtains 92.47 (0.49).
> However, rather than being competing methods, we believe that the cut statistic is *complementary* to methods like ASTRA and COSINE. We include more details below.

---

> > ### Author Response · Authors · 2022-08-02
> > **Combining ASTRA, COSINE with the cut statistic**
> >
> > You're absolutely right that COSINE and ASTRA incorporate some model-confidence-based data selection into the training procedure. However, we view self-training approaches like COSINE and ASTRA as somewhat orthogonal to the cut statistic: the techniques can be combined.
> >
> > For example, consider the first training step of ASTRA in each round: the student model is pretrained on a small amount of labeled data, and then the RAN teacher model is trained on *all* of the data using the student pseudolabels as a weak label.
> > Instead, we could train the teacher on weakly-labeled data selected by the cut statistic. This provides a higher-quality training set for the teacher model in each round of ASTRA.
> > Likewise, instead of training the student model on all of the data with pseudolabels from the teacher, we can still select a subset with the cut statistic (although this step is less necessary, since the teacher model already applies instance-specific weighting).
> > Our preliminary experiments show that adding the cut statistic to ASTRA in this way boosts the performance.
> >
> > For TREC we ran ASTRA with 20 randomly selected labeled examples for 10 different random seeds. The test accuracy improves from 73.36 (3.20) for base ASTRA to 75.27 (1.60) for ASTRA+cutstat. For 40 labeled examples, the improvement is from 83.89 (3.41) to 85.14 (2.71).
> > We performed the same experiment for SemEval. For 20 labeled examples, ASTRA obtains test accuracy 87.33 (1.23) for 10 random seeds. ASTRA + cutstat obtains test accuracy 90.35 (1.56). For 40 labeled examples, the improvement is from 88.48 (1.34) to 91.12 (1.16).
> > This shows that the cut statistic can also improve methods that do instance-specific-weighting during the end model training.
> >
> > The cut statistic can be integrated with COSINE in a similar way. First, we train the initial model on weakly-labeled data selected by the cut statistic, and then proceed with the COSINE self-training iterations. This provides a higher-quality initial model to kick off the self-training process for COSINE. We believe the core idea of ASTRA and COSINE is to use the unlabeled data, not to perform data selection. So we view the techniques as complementary, and combining them is an exciting direction for future research. Note that prior work has also shown that the cut statistic and self-training methods can be successfully combined (our reference [17]).
> >
> > In an updated version of the paper, we will include a more detailed discussion on how to incorporate the cut statistic into iterative approaches like ASTRA and COSINE.
> > We can also release code for ASTRA+cutstat, building on the WRENCH implementation of ASTRA. We chose to not discuss these combinations in the initial version of the paper for simplicity and because we wanted to focus on improving the most popular methods in practice.
> > Thanks for the great suggestion on applying this method to state-of-the-art joint approaches.

---

> > > ### Author Response · Authors · 2022-08-09
> > > **Did we address all of your concerns?**
> > >
> > > Thanks again for your helpful comments on our submission. We hope we addressed your concerns about better contextualizing our approach compared to recent state-of-the-art joint approaches to weak supervision like COSINE and ASTRA. We plan to include the COSINE results reported by Zhang et al. to better contextualize our numbers. But more excitingly, our preliminary experiments (described in "Combining ASTRA, COSINE with the cut statistic") indicate that COSINE/ASTRA can be effectively combined with the cut statistic to further boost performance, which is in keeping with prior work combining the cut statistic with self-training methods.
> > >
> > > Do you have any other concerns that we can help clear up?

---

### Meta-Review · Area_Chair_tAoo · 2022-08-26

**Recommendation:** Accept
**Confidence:** Certain

**Metareview:**

This paper examines the question of whether it is best to use all the available weakly labeled data to train a model. Contrary to usual practice, it finds that it is often best to filter that data. The creativity of the paper is using a statistic called the cut statistic to select high quality subset of the data. It uses a graph where nearest neighbors of examples are connected using the distance induced by some representation. The approach is very elegant. It can be plugged into many existing approaches. Experiments on the WRENCH benchmark for weak supervision show that this selection method consistently improves five different approaches to creating weak labels.

The reviewers all agreed that the paper makes a strong contribution, is clear and well written, and provides an interesting analysis of why the method works.

**Award:**

No

---

### Decision · Program_Chairs · 2022-09-14

Accept